# A Finite-Time Analysis of Q-Learning with Neural Network Function Approximation

## Abstract

Q-learning with neural network function approximation (neural Q-learning for short) is among the most prevalent deep reinforcement learning algorithms. Despite its empirical success, the non-asymptotic convergence rate of neural Q-learning remains virtually unknown. In this paper, we present a finite-time analysis of a neural Q-learning algorithm, where the data are generated from a Markov decision process and the action-value function is approximated by a deep ReLU neural network. We prove that neural Q-learning finds the optimal policy with $O(1/\sqrt{T})$ convergence rate if the neural function approximator is sufficiently overparameterized, where $T$ is the number of iterations. To our best knowledge, our result is the first finite-time analysis of neural Q-learning under non-i.i.d. data assumption.

## 1 Introduction

Q-learning has been shown to be one of the most important and effective learning strategies in Reinforcement Learning (RL) over the past decades (Watkins & Dayan, 1992; Schmidhuber, 2015; Sutton & Barto, 2018), where the agent takes an action based on the action-value function (a.k.a., Q-value function) at the current state. Recent advance in deep learning has also enabled the application of Q-learning algorithms to large-scale decision problems such as mastering Go (Silver et al., 2016; 2017), robotic motion control (Levine et al., 2015; Kalashnikov et al., 2018) and autonomous driving (Shalev-Shwartz et al., 2016; Schwarting et al., 2018). In particular, the seminal work by Mnih et al. (2015) introduced the Deep Q-Network (DQN) to approximate the action-value function and achieved a superior performance versus a human expert in playing Atari games, which triggers a line of research on deep reinforcement learning such as Double Deep Q-Learning (Van Hasselt et al., 2016) and Dueling DQN (Wang et al., 2016).

Apart from its widespread empirical success in numerous applications, the convergence of Q-learning and temporal difference (TD) learning algorithms has also been extensively studied in the literature (Jaakkola et al., 1994; Baird, 1995; Tsitsiklis & Van Roy, 1997; Perkins & Pendrith, 2002; Melo et al., 2008; Mehta & Meyn, 2009; Liu et al., 2015; Bhandari et al., 2018; Lakshminarayanan & Szepesvari, 2018; Zou et al., 2019b). However, the convergence guarantee of deep Q-learning algorithms remains a largely open problem. The only exceptions are Yang et al. (2019) which studied the fitted Q-iteration (FQI) algorithm (Riedmiller, 2005; Munos & Szepesvári, 2008) with action-value function approximation based on a sparse ReLU network, and Cai et al. (2019a) which studied the global convergence of Q-learning algorithm with an i.i.d. observation model and action-value function approximation based on a two-layer neural network. The main limitation of the aforementioned work is the unrealistic assumption that all the data used in the Q-learning algorithm are sampled i.i.d. from a fixed stationary distribution, which fails to capture the practical setting of neural Q-learning.

In this paper, in order to bridge the gap between the empirical success of neural Q-learning and the theory of conventional Q-learning (i.e., tabular Q-learning, and Q-learning with linear function approximation), we study the non-asymptotic convergence of a neural Q-learning algorithm under non-i.i.d. observations. In particular, we use a deep neural network with the ReLU activation function to approximate the action-value function. In each iteration of the neural Q-learning algorithm, it updates the network weight parameters using the temporal difference (TD) error and the gradient of the neural network function. Our work extends existing finite-time analyses for TD learning (Bhandari et al., 2018) and Q-learning (Zou et al., 2019b), from linear function approximation to deep

Table 1: Comparison with existing finite-time analyses of Q-learning.

| | Non-i.i.d. | Neural Approximation | Multiple Layers | Rate |
|---|:---:|:---:|:---:|:---:|
| Bhandari et al. (2018) | ✓ | ✗ | ✗ | $O(1/T)$ |
| Zou et al. (2019b) | ✓ | ✗ | ✗ | $O(1/T)$ |
| Cai et al. (2019a) | ✗ | ✓ | ✗ | $O(1/\sqrt{T})$ |
| This paper | ✓ | ✓ | ✓ | $O(1/\sqrt{T})$ |

neural network based function approximation. Compared with the very recent theoretical work for neural Q-learning (Yang et al., 2019; Cai et al., 2019a), our analysis relaxes the non-realistic i.i.d. data assumption and applies to neural network approximation with arbitrary number of layers. Our main contributions are summarized as follows

- We establish the first finite-time analysis of Q-learning with deep neural network function approximation when the data are generated from a Markov decision process (MDP). We show that, when the network is sufficiently wide, neural Q-learning converges to the optimal action-value function up to the approximation error of the neural network function class.

- We establish an $O(1/\sqrt{T})$ convergence rate of neural Q-learning to the optimal Q-value function up to the approximation error, where $T$ is the number of iterations. This convergence rate matches the one for TD-learning with linear function approximation and constant stepsize (Bhandari et al., 2018). Although we study a more challenging setting where the data are non-i.i.d. and the neural network approximator has multiple layers, our convergence rate also matches the $O(1/\sqrt{T})$ rate proved in Cai et al. (2019a) with i.i.d. data and a two-layer neural network approximator.

To sum up, we present a comprehensive comparison between our work and the most relevant work in terms of their respective settings and convergence rates in Table 1.

**Notation** We denote $[n] = \{1, \ldots, n\}$ for $n \in \mathbb{N}^+$. $\|\mathbf{x}\|_2$ is the Euclidean norm of a vector $\mathbf{x} \in \mathbb{R}^d$. For a matrix $\mathbf{W} \in \mathbb{R}^{m \times n}$, we denote by $\|\mathbf{W}\|_2$ and $\|\mathbf{W}\|_F$ its operator norm and Frobenius norm respectively. We denote by $\text{vec}(\mathbf{W})$ the vectorization of $\mathbf{W}$, which converts $\mathbf{W}$ into a column vector. For a semi-definite matrix $\mathbf{\Sigma} \in \mathbb{R}^{d \times d}$ and a vector $\mathbf{x} \in \mathbb{R}^d$, $\|\mathbf{x}\|_{\mathbf{\Sigma}} = \sqrt{\mathbf{x}^\top \mathbf{\Sigma} \mathbf{x}}$ denotes the Mahalanobis norm. We reserve the notations $\{C_i\}_{i=0,1,\ldots}$ to represent universal positive constants that are independent of problem parameters. The specific value of $\{C_i\}_{i=1,2,\ldots}$ can be different line by line. We write $a_n = O(b_n)$ if $a_n \leq C b_n$ for some constant $C > 0$ and $a_n = \widetilde{O}(b_n)$ if $a_n = O(b_n)$ up to some logarithmic terms of $b_n$.

## 2 RELATED WORK

Due to the huge volume of work in the literature for TD learning and Q-learning algorithms, we only review the most relevant work here.

**Asymptotic analysis** The asymptotic convergence of TD learning and Q-learning algorithms has been well established in the literature (Jaakkola et al., 1994; Tsitsiklis & Van Roy, 1997; Konda & Tsitsiklis, 2000; Borkar & Meyn, 2000; Ormoneit & Sen, 2002; Melo et al., 2008; Devraj & Meyn, 2017). In particular, Tsitsiklis & Van Roy (1997) specified the precise conditions for TD learning with linear function approximation to converge and gave counterexamples that diverge. Melo et al. (2008) proved the asymptotic convergence of Q-learning with linear function approximation from standard ODE analysis, and identified a critic condition on the relationship between the learning policy and the greedy policy that ensures the almost sure convergence.

**Finite-time analysis** The finite-time analysis of the convergence rate for Q-learning algorithms has been largely unexplored until recently. In specific, Dalal et al. (2018); Lakshminarayanan & Szepesvari (2018) studied the convergence of TD(0) algorithm with linear function approximation under i.i.d. data assumptions and constant step sizes. Concurrently, a seminal work by Bhandari et al. (2018) provided a unified framework of analysis for TD learning under both i.i.d. and Markovian noise assumptions with an extra projection step. The analysis has been extended by Zou et al. (2019b) to SARSA and Q-learning algorithms with linear function approximation. More recently,

Srikant & Ying (2019) established the finite-time convergence for TD learning algorithms with linear function approximation and a constant step-size without the extra projection step under non-i.i.d. data assumptions through carefully choosing the Lyapunov function for the associated ordinary differential equation of TD update. A similar analysis was also extended to Q-learning with linear function approximation (Chen et al., 2019). Hu & Syed (2019) further provided a unified analysis for a class of TD learning algorithms using Markov jump linear system.

**Neural function approximation** Despite the empirical success of DQN, the theoretical convergence of Q-learning with deep neural network approximation is still missing in the literature. Following the recent advances in the theory of deep learning for overparameterized networks (Jacot et al., 2018; Chizat & Bach, 2018; Du et al., 2019b;a; Allen-Zhu et al., 2019b;a; Zou et al., 2019a; Arora et al., 2019; Cao & Gu, 2019a; Zou & Gu, 2019; Cai et al., 2019b), two recent work by Yang et al. (2019) and Cai et al. (2019a) proved the convergence rates of fitted Q-iteration and Q-learning with a sparse multi-layer ReLU network and two-layer neural network approximation respectively, under i.i.d. observations.

## 3 PRELIMINARIES

A discrete-time Markov Decision Process (MDP) is denoted by a tuple $\mathcal{M} = (\mathcal{S}, \mathcal{A}, \mathcal{P}, r, \gamma)$. $\mathcal{S}$ and $\mathcal{A}$ are the sets of all states and actions respectively. $\mathcal{P} : \mathcal{S} \times \mathcal{A} \to \mathcal{P}(\mathcal{S})$ is the transition kernel such that $\mathcal{P}(s'|s,a)$ gives the probability of transiting to state $s'$ after taking action $a$ at state $s$. $r : \mathcal{S} \times \mathcal{A} \to [-1,1]$ is a deterministic reward function. $\gamma \in (0,1)$ is the discounted factor. A policy $\pi : \mathcal{S} \to \mathcal{P}(\mathcal{A})$ is a function mapping a state $s \in \mathcal{S}$ to a probability distribution $\pi(\cdot|s)$ over the action space. Let $s_t$ and $a_t$ denote the state and action at time step $t$. Then the transition kernel $\mathcal{P}$ and the policy $\pi$ determine a Markov chain $\{s_t\}_{t=0,1,\ldots}$ For any fixed policy $\pi$, its associated value function $V^\pi : \mathcal{S} \to \mathbb{R}$ is defined as the expected total discounted reward:

$$V^\pi(s) = \mathbb{E}[\textstyle\sum_{t=0}^\infty \gamma^t r(s_t, a_t)|s_0 = s], \quad \forall s \in \mathcal{S}.$$

The corresponding action-value function $Q^\pi : \mathcal{S} \times \mathcal{A} \to \mathbb{R}$ is defined as

$$Q^\pi(s,a) = \mathbb{E}[\textstyle\sum_{t=0}^\infty \gamma^t r(s_t, a_t)|s_0 = s, a_0 = a] = r(s,a) + \gamma \int_{\mathcal{S}} V^\pi(s')\mathcal{P}(s'|s,a)\mathrm{d}s',$$

for all $s \in \mathcal{S}, a \in \mathcal{A}$. The optimal action-value function $Q^*$ is defined as $Q^*(s,a) = \sup_\pi Q^\pi(s,a)$ for all $(s,a) \in \mathcal{S} \times \mathcal{A}$. Based on $Q^*$, the optimal policy $\pi^*$ can be derived by following the greedy algorithm such that $\pi^*(a|s) = 1$ if $Q(s,a) = \max_{b \in \mathcal{A}} Q^*(s,b)$ and $\pi^*(a|s) = 0$ otherwise. We define the optimal Bellman operator $\mathcal{T}$ as follows

$$\mathcal{T}Q(s,a) = r(s,a) + \gamma \cdot \mathbb{E}\big[\max_{b \in \mathcal{A}} Q(s',b)|s' \sim \mathcal{P}(\cdot|s,a)\big]. \tag{3.1}$$

It is worth noting that the optimal Bellman operator $\mathcal{T}$ is $\gamma$-contractive in the sup-norm and $Q^*$ is the unique fixed point of $\mathcal{T}$ (Bertsekas et al., 1995).

## 4 THE NEURAL Q-LEARNING ALGORITHM

In this section, we start with a brief review of Q-learning with linear function approximation. Then we will present the neural Q-learning algorithm.

### 4.1 Q-LEARNING WITH LINEAR FUNCTION APPROXIMATION

In many reinforcement learning algorithms, the goal is to estimate the action-value function $Q(\cdot,\cdot)$, which can be formulated as minimizing the mean-squared Bellman error (MSBE) (Sutton & Barto, 2018):

$$\min_{Q(\cdot,\cdot)} \mathbb{E}_{\mu,\pi,\mathcal{P}}\big[(\mathcal{T}Q(s,a) - Q(s,a))^2\big], \tag{4.1}$$

where state $s$ is generated from the initial state distribution $\mu$ and action $a$ is chosen based on a fixed learning policy $\pi$. To optimize (4.1), Q-learning iteratively updates the action-value function using the Bellman operator in (3.1), i.e., $Q_{t+1}(s,a) = \mathcal{T}Q_t(s,a)$ for all $(s,a) \in \mathcal{S} \times \mathcal{A}$. However, due to the large state and action spaces, whose cardinalities, i.e., $|\mathcal{S}|$ and $|\mathcal{A}|$, can be infinite for continuous problems in many applications, the aforementioned update is impractical. To address this

issue, a linear function approximator is often used (Szepesvari, 2010; Sutton & Barto, 2018), where the action-value function is assumed to be parameterized by a linear function, i.e., $Q(s, a; \boldsymbol{\theta}) = \phi(s, a)^\top \boldsymbol{\theta}$ for any $(s, a) \in \mathcal{S} \times \mathcal{A}$, where $\phi : \mathcal{S} \times \mathcal{A} \to \mathbb{R}^d$ maps the state-action pair to a $d$-dimensional vector, and $\boldsymbol{\theta} \in \boldsymbol{\Theta} \subseteq \mathbb{R}^d$ is an unknown weight vector. The minimization problem in (4.1) then turns to minimizing the MSBE over the parameter space $\boldsymbol{\Theta}$.

## 4.2 NEURAL Q-LEARNING

Analogous to Q-learning with linear function approximation, the action-value function can also be approximated by a deep neural network to increase the representation power of the approximator. Specifically, we define a $L$-hidden-layer neural network as follows

$$f(\boldsymbol{\theta}; \mathbf{x}) = \sqrt{m} \mathbf{W}_L \sigma_L(\mathbf{W}_{L-1} \cdots \sigma(\mathbf{W}_1 \mathbf{x}) \cdots), \tag{4.2}$$

where $\mathbf{x} \in \mathbb{R}^d$ is the input data, $\mathbf{W}_1 \in \mathbb{R}^{m \times d}$, $\mathbf{W}_L \in \mathbb{R}^{1 \times m}$ and $\mathbf{W}_l \in \mathbb{R}^{m \times m}$ for $l = 2, \ldots, L - 1$, $\boldsymbol{\theta} = (\mathrm{vec}(\mathbf{W}_1)^\top, \ldots, \mathrm{vec}(\mathbf{W}_L)^\top)^\top$ is the concatenation of the vectorization of all parameter matrices, and $\sigma(x) = \max\{0, x\}$ is the ReLU activation function. Then, we can parameterize $Q(s, a)$ using a deep neural network as $Q(s, a; \boldsymbol{\theta}) = f(\boldsymbol{\theta}; \phi(s, a))$, where $\boldsymbol{\theta} \in \boldsymbol{\Theta}$ and $\phi : \mathcal{S} \times \mathcal{A} \to \mathbb{R}^d$ is a feature mapping. Without loss of generality, we assume that $\|\phi(s, a)\|_2 \leq 1$ in this paper. Let $\pi$ be an arbitrarily stationary policy. The MSBE minimization problem in (4.1) can be rewritten in the following form

$$\min_{\boldsymbol{\theta} \in \boldsymbol{\Theta}} \mathbb{E}_{\mu, \pi, \mathcal{P}} \big[ (Q(s, a; \boldsymbol{\theta}) - \mathcal{T} Q(s, a; \boldsymbol{\theta}))^2 \big]. \tag{4.3}$$

Recall that the optimal action-value function $Q^*$ is the fixed point of Bellman optimality operator $\mathcal{T}$ which is $\gamma$-contractive. Therefore $Q^*$ is the unique global minimizer of (4.3).

The nonlinear parameterization of $Q(\cdot, \cdot)$ turns the MSBE in (4.3) to be highly nonconvex, which imposes difficulty in finding the global optimum $\boldsymbol{\theta}^*$. To mitigate this issue, we will approximate the solution of (4.3) by project the Q-value function into some function class parameterized by $\boldsymbol{\theta}$, which leads to minimizing the mean square projected Bellman error (MSPBE):

$$\min_{\boldsymbol{\theta} \in \boldsymbol{\Theta}} \mathbb{E}_{\mu, \pi, \mathcal{P}} \big[ (Q(s, a; \boldsymbol{\theta}) - \Pi_{\mathcal{F}} \mathcal{T} Q(s, a; \boldsymbol{\theta}))^2 \big], \tag{4.4}$$

where $\mathcal{F} = \{Q(\cdot, \cdot; \boldsymbol{\theta}) : \boldsymbol{\theta} \in \boldsymbol{\Theta}\}$ is some function class parameterized by $\boldsymbol{\theta} \in \boldsymbol{\Theta}$, and $\Pi_{\mathcal{F}}$ is a projection operator. Then the neural Q-learning algorithm updates the weight parameter $\boldsymbol{\theta}$ using the following projected descent step: $\boldsymbol{\theta}_{t+1} = \Pi_{\boldsymbol{\Theta}}(\boldsymbol{\theta}_t - \eta_t \mathbf{g}_t(\boldsymbol{\theta}_t))$, where the gradient term $\mathbf{g}_t(\boldsymbol{\theta}_t)$ is defined as

$$\mathbf{g}_t(\boldsymbol{\theta}_t) = \nabla_{\boldsymbol{\theta}} f(\boldsymbol{\theta}_t; \phi(s_t, a_t)) \big( f(\boldsymbol{\theta}_t; \phi(s_t, a_t)) - r_t - \gamma \max_{b \in \mathcal{A}} f(\boldsymbol{\theta}_t; \phi(s_{t+1}, b)) \big)$$

$$\overset{\text{def}}{=} \Delta_t(s_t, a_t, s_{t+1}; \boldsymbol{\theta}_t) \nabla_{\boldsymbol{\theta}} f(\boldsymbol{\theta}_t; \phi(s_t, a_t)), \tag{4.5}$$

and $\Delta_t$ is the temporal difference (TD) error. It should be noted that $\mathbf{g}_t$ is not the gradient of the MSPBE nor an unbiased estimator for it. The details of the neural Q-learning algorithm are displayed in Algorithm 1, where $\boldsymbol{\theta}_0$ is randomly initialized, and the constraint set is chosen to be $\boldsymbol{\Theta} = \mathbb{B}(\boldsymbol{\theta}_0, \omega)$, which is defined as follows

$$\mathbb{B}(\boldsymbol{\theta}_0, \omega) \overset{\text{def}}{=} \{\boldsymbol{\theta} = (\mathrm{vec}(\mathbf{W}_1)^\top, \ldots, \mathrm{vec}(\mathbf{W}_L)^\top)^\top : \|\mathbf{W}_l - \mathbf{W}_l^{(0)}\|_F \leq \omega, l = 1, \ldots, L\} \tag{4.6}$$

for some tunable parameter $\omega$. It is easy to verify that $\|\boldsymbol{\theta} - \boldsymbol{\theta}'\|_2^2 = \sum_{l=1}^L \|\mathbf{W}_l - \mathbf{W}_l'\|_F^2$.

## 5 CONVERGENCE ANALYSIS OF NEURAL Q-LEARNING

In this section, we provide a finite-sample analysis of neural Q-learning. Note that the optimization problem in (4.4) is nonconvex. We focus on finding a surrogate action-value function in the neural network function class that well approximates $Q^*$.

### 5.1 APPROXIMATE STATIONARY POINT IN THE CONSTRAINED SPACE

To ease the presentation, we abbreviate $f(\boldsymbol{\theta}; \phi(s, a))$ as $f(\boldsymbol{\theta})$ when no confusion arises. We define the function class $\mathcal{F}_{\boldsymbol{\Theta}, m}$ as a collection of all local linearization of $f(\boldsymbol{\theta})$ at the initial point $\boldsymbol{\theta}_0$

$$\mathcal{F}_{\boldsymbol{\Theta}, m} = \{f(\boldsymbol{\theta}_0) + \langle \nabla_{\boldsymbol{\theta}} f(\boldsymbol{\theta}_0), \boldsymbol{\theta} - \boldsymbol{\theta}_0 \rangle : \boldsymbol{\theta} \in \boldsymbol{\Theta}\}. \tag{5.1}$$

---

**Algorithm 1** Neural Q-Learning with Gaussian Initialization

---

1: **Input:** learning policy $\pi$, learning rate $\{\eta_t\}_{t=0,1,\dots}$, discount factor $\gamma$, constraint set $\boldsymbol{\Theta}$, Randomly generate the entries of $\mathbf{W}_l^{(0)}$ from $N(0, 1/m)$, $l = 1, \dots, m$
2: **Initialization:** $\boldsymbol{\theta}_0 = (\mathbf{W}_0^{(1)\top}, \dots, \mathbf{W}_0^{(L)\top})^\top$
3: **for** $t = 0, \dots, T - 1$ **do**
4:     Sample data $(s_t, a_t, r_t, s_{t+1})$ from policy $\pi$
5:     $\Delta_t = f(\boldsymbol{\theta}_t; \phi(s_t, a_t)) - (r_t + \gamma \max_{b \in \mathcal{A}} f(\boldsymbol{\theta}_t; \phi(s_{t+1}, b)))$
6:     $\mathbf{g}_t(\boldsymbol{\theta}_t) = \nabla_{\boldsymbol{\theta}} f(\boldsymbol{\theta}_t; \phi(s_t, a_t)) \Delta_t$
7:     $\boldsymbol{\theta}_{t+1} = \Pi_{\boldsymbol{\Theta}}(\boldsymbol{\theta}_t - \eta_t \mathbf{g}_t(\boldsymbol{\theta}_t))$
8: **end for**

---

Following to the local linearization analysis in Cai et al. (2019a), we define the approximate stationary point of Algorithm 1 as follows.

**Definition 5.1** (Cai et al. (2019a)). A point $\boldsymbol{\theta}^* \in \boldsymbol{\Theta}$ is said to be the approximate stationary point of Algorithm 1 if for all $\boldsymbol{\theta} \in \boldsymbol{\Theta}$ it holds that

$$\mathbb{E}_{\mu,\pi,\mathcal{P}}\big[\widehat{\Delta}(s, a, s'; \boldsymbol{\theta}^*) \langle \nabla_{\boldsymbol{\theta}} \widehat{f}(\boldsymbol{\theta}^*; \phi(s, a)), \boldsymbol{\theta} - \boldsymbol{\theta}^* \rangle\big] \geq 0, \tag{5.2}$$

where $\widehat{f}(\boldsymbol{\theta}; \phi(s, a)) := \widehat{f}(\boldsymbol{\theta}) \in \mathcal{F}_{\boldsymbol{\Theta},m}$ and the temporal difference error $\widehat{\Delta}$ is

$$\widehat{\Delta}(s, a, s'; \boldsymbol{\theta}) = \widehat{f}(\boldsymbol{\theta}; \phi(s, a)) - \big(r(s, a) + \gamma \max_{b \in \mathcal{A}} \widehat{f}(\boldsymbol{\theta}; \phi(s', b))\big). \tag{5.3}$$

For any $\widehat{f} \in \mathcal{F}_{\boldsymbol{\Theta},m}$, it holds that $\langle \nabla_{\boldsymbol{\theta}} \widehat{f}(\boldsymbol{\theta}^*), \boldsymbol{\theta} - \boldsymbol{\theta}^* \rangle = \langle \nabla_{\boldsymbol{\theta}} f(\boldsymbol{\theta}_0), \boldsymbol{\theta} - \boldsymbol{\theta}^* \rangle = \widehat{f}(\boldsymbol{\theta}) - \widehat{f}(\boldsymbol{\theta}^*)$. Definition 5.1 immediately implies

$$\mathbb{E}_{\mu,\pi,\mathcal{P}}\big[\big(\widehat{f}(\boldsymbol{\theta}^*) - \mathcal{T}\widehat{f}(\boldsymbol{\theta}^*)\big)\big(\widehat{f}(\boldsymbol{\theta}) - \widehat{f}(\boldsymbol{\theta}^*)\big)\big] \geq 0, \qquad \forall \boldsymbol{\theta} \in \boldsymbol{\Theta}. \tag{5.4}$$

According to Proposition 4.2 in Cai et al. (2019a), this further indicates $\widehat{f}(\boldsymbol{\theta}^*) = \Pi_{\mathcal{F}_{\boldsymbol{\Theta},m}} \mathcal{T}\widehat{f}(\boldsymbol{\theta}^*)$. In other words, $\widehat{f}(\boldsymbol{\theta}^*)$ is the unique fixed point of the MSPBE in (4.4). Therefore, we can show the convergence of neural Q-learning to the optimal action-value function $Q^*$ by first connecting it to the minimizer $\widehat{f}(\boldsymbol{\theta}^*)$ and then adding the approximation error of $\mathcal{F}_{\boldsymbol{\Theta},m}$.

## 5.2 THE MAIN THEORY

Before we present the convergence of Algorithm 1, let us lay down the assumptions used throughout our paper. The first assumption controls the bias caused by the Markovian noise in the observations through assuming the uniform ergodicity of the Markov chain generated by the learning policy $\pi$.

**Assumption 5.2.** The learning policy $\pi$ and the transition kernel $\mathcal{P}$ induce a Markov chain $\{s_t\}_{t=0,1,\dots}$ such that there exist constants $\lambda > 0$ and $\rho \in (0, 1)$ satisfying

$$\sup_{s \in \mathcal{S}} d_{TV}(\mathbb{P}(s_t \in \cdot | s_0 = s), \pi) \leq \lambda \rho^t, \quad \text{for all } t = 0, 1, \dots$$

Assumption 5.2 also appears in Bhandari et al. (2018); Zou et al. (2019b), which is essential for the analysis of the Markov decision process. The uniform ergodicity can be established via the minorization condition for irreducible Markov chains (Meyn & Tweedie, 2012; Levin & Peres, 2017).

For the purpose of exploration, we also need to assume that the learning policy $\pi$ satisfies some regularity condition. Denote $b_{\max}(\boldsymbol{\theta}) = \operatorname{argmax}_{b \in \mathcal{A}} |\langle \nabla_{\boldsymbol{\theta}} f(\boldsymbol{\theta}_0; s, b), \boldsymbol{\theta} \rangle|$ for any $\boldsymbol{\theta} \in \boldsymbol{\Theta}$. Similar to Melo et al. (2008); Zou et al. (2019b); Chen et al. (2019), we define

$$\boldsymbol{\Sigma}_\pi = 1/m \mathbb{E}_{\mu,\pi}\big[\nabla_{\boldsymbol{\theta}} f(\boldsymbol{\theta}_0; s, a) \nabla_{\boldsymbol{\theta}} f(\boldsymbol{\theta}_0; s, a)^\top\big], \tag{5.5}$$

$$\boldsymbol{\Sigma}_\pi^*(\boldsymbol{\theta}) = 1/m \mathbb{E}_{\mu,\pi}\big[\nabla_{\boldsymbol{\theta}} f(\boldsymbol{\theta}_0; s, b_{\max}(\boldsymbol{\theta})) \nabla_{\boldsymbol{\theta}} f(\boldsymbol{\theta}_0; s, b_{\max}(\boldsymbol{\theta}))^\top\big]. \tag{5.6}$$

Note that $\boldsymbol{\Sigma}_\pi$ is independent of $\boldsymbol{\theta}$ and only depends on the policy $\pi$ and the initial point $\boldsymbol{\theta}_0$ in the definition of $\widehat{f}$. In contrast, $\boldsymbol{\Sigma}_\pi^*(\boldsymbol{\theta})$ is defined based on the greedy action under the policy associated with $\boldsymbol{\theta}$. The scaling parameter $1/m$ is used to ensure that the operator norm of $\boldsymbol{\Sigma}_\pi$ to be in the order of $O(1)$. It is worth noting that $\boldsymbol{\Sigma}_\pi$ is different from the neural tangent kernel (NTK) or the Gram matrix in Jacot et al. (2018); Du et al. (2019a); Arora et al. (2019), which are $n \times n$ matrices defined based on a finite set of data points $\{(s_i, a_i)\}_{i=1,\dots,n}$. When $f$ is linear, $\boldsymbol{\Sigma}_\pi$ reduces to the covariance matrix of the feature vector.

**Assumption 5.3.** There exists a constant $\alpha > 1$ such that $\Sigma_\pi - \alpha\gamma^2\Sigma_\pi^*(\boldsymbol{\theta}) \succ \mathbf{0}$ for all $\boldsymbol{\theta}$ and $\boldsymbol{\theta}_0$.

Assumption 5.3 is also made for Q-learning with linear function approximation in Melo et al. (2008); Zou et al. (2019b); Chen et al. (2019). Moreover, Chen et al. (2019) presented numerical simulations to verify the validity of Assumption 5.3. Cai et al. (2019a) imposed a slightly different assumption but with the same idea that the learning policy $\pi$ should be not too far away from the greedy policy. The regularity assumption on the learning policy is directly imposed on the action value function in Cai et al. (2019a), which can be implied by Assumption 5.3 and thus is slightly weaker. We note that Assumption 5.3 can be relaxed to the one made in Cai et al. (2019a) without changing any of our analysis. Nevertheless, we choose to present the current version which is more consistent with existing work on Q-learning with linear function approximation (Melo et al., 2008; Chen et al., 2019).

**Theorem 5.4.** Suppose Assumptions 5.2 and 5.3 hold. The constraint set $\Theta$ is defined as in (4.6). We set the radius as $\omega = C_0 m^{-1/2} L^{-9/4}$, the step size in Algorithm 1 as $\eta = 1/(2(1-\alpha^{-1/2})mT)$, and the width of the neural network as $m \geq C_1 \max\{dL^2\log(m/\delta), \omega^{-4/3}L^{-8/3}\log(m/(\omega\delta))\}$, where $\delta \in (0,1)$. Then with probability at least $1 - 2\delta - L^2\exp(-C_2 m^{2/3}L)$ over the randomness of the Gaussian initialization $\boldsymbol{\theta}_0$, it holds that

$$\frac{1}{T}\sum_{t=0}^{T-1}\mathbb{E}\big[(\widehat{f}(\boldsymbol{\theta}_t) - \widehat{f}(\boldsymbol{\theta}^*))^2\big|\boldsymbol{\theta}_0\big] \leq \frac{1}{\sqrt{T}} + \frac{C_2\tau^*\log(T/\delta)\log T}{\beta^2\sqrt{T}} + \frac{C_3\log m\log(T/\delta)}{\beta m^{1/6}},$$

where $\beta = 1 - \alpha^{-1/2} \in (0,1)$ is a constant, $\tau^* = \min\{t = 0,1,2,\ldots|\lambda\rho^t \leq \eta_T\}$ is the mixing time of the Markov chain $\{s_t, a_t\}_{t=0,1,\ldots}$, and $\{C_i\}_{i=0,\ldots,5}$ are universal constants independent of problem parameters.

**Remark 5.5.** Theorem 5.4 characterizes the distance between the output of Algorithm 1 to the approximate stationary point defined in function class $\mathcal{F}_{\Theta,m}$. From (5.4), we know that $\widehat{f}(\boldsymbol{\theta}^*)$ is the minimizer of the MSPBE (4.4). Note that $\tau^*$ is in the order of $O(\log(mT/\log T))$. Theorem 5.4 suggests that neural Q-learning converges to the minimizer of MSPBE with a rate in the order of $O((\log(mT))^3/\sqrt{T} + \log m\log T/m^{1/6})$, which reduces to $\widetilde{O}(1/\sqrt{T})$ when the width $m$ of the neural network is sufficiently large.

In the following theorem, we show that neural Q-learning converges to the optimal action-value function within finite time if the neural network is overparameterized.

**Theorem 5.6.** Under the same conditions as in Theorem 5.4, with probability at least $1 - 3\delta - L^2\exp(-C_0 m^{2/3}L)$ over the randomness of $\boldsymbol{\theta}_0$, it holds that

$$\frac{1}{T}\sum_{t=0}^{T-1}\mathbb{E}\big[(Q(s,a;\boldsymbol{\theta}_t) - Q^*(s,a))^2\big] \leq \frac{3\mathbb{E}\big[\big(\Pi_{\mathcal{F}_{\Theta,m}}Q^*(s,a) - Q^*(s,a)\big)^2\big]}{(1-\gamma)^2} + \frac{1}{\sqrt{T}}$$
$$+ \frac{C_1\tau^*\log(T/\delta)\log T}{\beta^2\sqrt{T}} + \frac{C_2\log(T/\delta)\log m}{\beta m^{1/6}},$$

where all the expectations are taken conditional on $\boldsymbol{\theta}_0$, $Q^*$ is the optimal action-value function, $\delta \in (0,1)$ and $\{C_i\}_{i=0,\ldots,2}$ are universal constants.

The optimal policy $\pi^*$ can be obtained by the greedy algorithm derived based on $Q^*$.

**Remark 5.7.** The convergence rate in Theorem 5.6 can be simplifies as follows

$$\frac{1}{T}\sum_{t=0}^{T-1}\mathbb{E}[(Q(s,a;\boldsymbol{\theta}_t) - Q^*(s,a))^2\big|\boldsymbol{\theta}_0] = \widetilde{O}\bigg(\mathbb{E}\big[\big(\Pi_{\mathcal{F}_{\Theta,m}}Q^*(s,a) - Q^*(s,a)\big)^2\big] + \frac{1}{m^{1/6}} + \frac{1}{\sqrt{T}}\bigg).$$

The first term is the projection error of the optimal Q-value function on to the function class $\mathcal{F}_{\Theta,m}$, which decreases to zero as the representation power of $\mathcal{F}_{\Theta,m}$ increases. In fact, when the width $m$ of the DNN is sufficiently large, recent studies (Cao & Gu, 2019a;b) show that $f(\boldsymbol{\theta})$ is almost linear around the initialization and the approximate stationary point $\widehat{f}(\boldsymbol{\theta}^*)$ becomes the fixed solution of the MSBE (Cai et al., 2019a). Moreover, this term diminishes when the $Q$ function is approximated by linear functions when the underlying parameter has a bounded norm (Bhandari et al., 2018; Zou et al., 2019b). As $m$ goes to infinity, we obtain the convergence of neural Q-learning to the optimal Q-value function with an $O(1/\sqrt{T})$ rate.

# 6 Proof of Main Results

In this section, we provide the detailed proof of the convergence of Algorithm 1. To simplify the presentation, we write $f(\boldsymbol{\theta}; \phi(s, a))$ as $f(\boldsymbol{\theta}; s, a)$ throughout the proof when no confusion arises.

We first define some notations that will simplify the presentation of the proof. Recall the definition of $\mathbf{g}_t(\cdot)$ in (4.5). For any $\boldsymbol{\theta} \in \boldsymbol{\Theta}$, we define the following vector-value map $\overline{\mathbf{g}}$ that is independent of the data point.

$$\overline{\mathbf{g}}(\boldsymbol{\theta}) = \mathbb{E}_{\mu,\pi,\mathcal{P}}[\nabla_{\boldsymbol{\theta}} f(\boldsymbol{\theta}; s, a)(f(\boldsymbol{\theta}; s, a) - r(s, a) - \gamma \max_{b \in \mathcal{A}} f(\boldsymbol{\theta}; s', b))], \tag{6.1}$$

where $s$ follows the initial state distribution $\mu$, $a$ is chosen based on the policy $\pi(\cdot|s)$ and $s'$ follows the transition probability $\mathcal{P}(\cdot|s, a)$. Similarly, for all $\boldsymbol{\theta} \in \boldsymbol{\Theta}$, we define the following gradient terms based on the linearized function $\widehat{f} \in \mathcal{F}_{\boldsymbol{\Theta}, m}$

$$\mathbf{m}_t(\boldsymbol{\theta}) = \widehat{\Delta}(s_t, a_t, s_{t+1}; \boldsymbol{\theta}) \nabla_{\boldsymbol{\theta}} \widehat{f}(\boldsymbol{\theta}), \quad \overline{\mathbf{m}}(\boldsymbol{\theta}) = \mathbb{E}_{\mu,\pi,\mathbf{P}}[\widehat{\Delta}(s, a, s'; \boldsymbol{\theta}) \nabla_{\boldsymbol{\theta}} \widehat{f}(\boldsymbol{\theta})], \tag{6.2}$$

where $\widehat{\Delta}$ is defined in (5.3), and a population version based on the linearized function.

Now we present the technical lemmas that are useful in our proof of Theorem 5.4. For the gradients $\mathbf{g}_t(\cdot)$ defined in (4.5) and $\mathbf{m}_t(\cdot)$ defined in (6.2), we have the following lemma that characterizes the difference between the gradient of the neural network function $f$ and the gradient of the linearized function $\widehat{f}$.

**Lemma 6.1.** The gradient of neural network function is close to the linearized gradient. Specifically, if $\boldsymbol{\theta}_t \in \mathbb{B}(\boldsymbol{\Theta}, \omega)$ and $m$ and $\omega$ satisfy

$$\begin{aligned} & m \geq C_0 \max\{dL^2 \log(m/\delta), \omega^{-4/3} L^{-8/3} \log(m/(\omega\delta))\}, \\ & \text{and} \quad C_1 d^{3/2} L^{-1} m^{-3/4} \leq \omega \leq C_2 L^{-6} (\log m)^{-3}, \end{aligned} \tag{6.3}$$

then it holds that

$$\begin{aligned} |\langle \mathbf{g}_t(\boldsymbol{\theta}_t) - \mathbf{m}_t(\boldsymbol{\theta}_t), \boldsymbol{\theta}_t - \boldsymbol{\theta}^* \rangle| \leq {} & C_3(2 + \gamma) \omega^{1/3} L^3 \sqrt{m \log m \log(T/\delta)} \|\boldsymbol{\theta}_t - \boldsymbol{\theta}^*\|_2 \\ & + (C_4 \omega^{4/3} L^{11/3} m \sqrt{\log m} + C_5 \omega^2 L^4 m) \|\boldsymbol{\theta}_t - \boldsymbol{\theta}^*\|_2, \end{aligned}$$

with probability at least $1 - 2\delta - 3L^2 \exp(-C_6 m \omega^{2/3} L)$ over the randomness of the initial point, and $\|\mathbf{g}_t(\boldsymbol{\theta}_t)\|_2 \leq (2 + \gamma) C_7 \sqrt{m \log(T/\delta)}$ holds with probability at least $1 - \delta - L^2 \exp(-C_6 m \omega^{2/3} L)$. where $\{C_i > 0\}_{i=0,\dots,7}$ are universal constants.

The next lemma upper bounds the bias of the non-i.i.d. data for the linearized gradient map.

**Lemma 6.2.** Suppose the step size sequence $\{\eta_0, \eta_1, \dots, \eta_T\}$ is nonincreasing. Then it holds that

$$\mathbb{E}[\langle \mathbf{m}_t(\boldsymbol{\theta}_t) - \overline{\mathbf{m}}(\boldsymbol{\theta}_t), \boldsymbol{\theta}_t - \boldsymbol{\theta}^* \rangle | \boldsymbol{\theta}_0] \leq C_0(m \log(T/\delta) + m^2 \omega^2) \tau^* \eta_{\max\{0, t-\tau^*\}},$$

for any fixed $t \leq T$, where $C_0 > 0$ is an universal constant and $\tau^* = \min\{t = 0, 1, 2, \dots | \lambda \rho^t \leq \eta_T\}$ is the mixing time of the Markov chain $\{s_t, a_t\}_{t=0,1,\dots}$.

Since $\widehat{f}$ is a linear function approximator of the neural network function $f$, we can show that the gradient of $\widehat{f}$ satisfies the following nice property in the constrained set $\boldsymbol{\Theta}$.

**Lemma 6.3.** Under Assumption 5.3, $\overline{\mathbf{m}}(\cdot)$ defined in (6.2) satisfies

$$\langle \overline{\mathbf{m}}(\boldsymbol{\theta}) - \overline{\mathbf{m}}(\boldsymbol{\theta}^*), \boldsymbol{\theta} - \boldsymbol{\theta}^* \rangle \geq (1 - \alpha^{-1/2}) \mathbb{E}[(\widehat{f}(\boldsymbol{\theta}) - \widehat{f}(\boldsymbol{\theta}^*))^2 | \boldsymbol{\theta}_0], \quad \forall \boldsymbol{\theta} \in \boldsymbol{\Theta}.$$

Now we can integrate the above results and obtain proof of Theorem 5.4.

*Proof of Theorem 5.4.* By Algorithm 1 and the non-expansiveness of projection $\Pi_{\boldsymbol{\Theta}}$, we have

$$\begin{aligned} \|\boldsymbol{\theta}_{t+1} - \boldsymbol{\theta}^*\|_2^2 &= \|\Pi_{\boldsymbol{\Theta}}(\boldsymbol{\theta}_t - \eta_t \mathbf{g}_t) - \boldsymbol{\theta}^*\|_2^2 \\ &\leq \|\boldsymbol{\theta}_t - \eta_t \mathbf{g}_t - \boldsymbol{\theta}^*\|_2^2 \\ &= \|\boldsymbol{\theta}_t - \boldsymbol{\theta}^*\|_2^2 + \eta_t^2 \|\mathbf{g}_t\|_2^2 - 2\eta_t \langle \mathbf{g}_t, \boldsymbol{\theta}_t - \boldsymbol{\theta}^* \rangle. \end{aligned} \tag{6.4}$$

We need to find an upper bound for the gradient norm and a lower bound for the inner product. According to Definition 5.1, the approximate stationary point $\boldsymbol{\theta}^*$ of Algorithm 1 satisfies $\langle \overline{\mathbf{m}}(\boldsymbol{\theta}^*), \boldsymbol{\theta} - \boldsymbol{\theta}^* \rangle \geq 0$ for all $\boldsymbol{\theta} \in \boldsymbol{\Theta}$. The inner product in (6.4) can be decomposed into

$$
\begin{aligned}
\langle \mathbf{g}_t, \boldsymbol{\theta}_t - \boldsymbol{\theta}^* \rangle &= \langle \mathbf{g}_t - \mathbf{m}_t(\boldsymbol{\theta}_t), \boldsymbol{\theta}_t - \boldsymbol{\theta}^* \rangle + \langle \mathbf{m}_t(\boldsymbol{\theta}_t) - \overline{\mathbf{m}}(\boldsymbol{\theta}_t), \boldsymbol{\theta}_t - \boldsymbol{\theta}^* \rangle + \langle \overline{\mathbf{m}}(\boldsymbol{\theta}_t), \boldsymbol{\theta}_t - \boldsymbol{\theta}^* \rangle \\
&\geq \langle \mathbf{g}_t - \mathbf{m}_t(\boldsymbol{\theta}_t), \boldsymbol{\theta}_t - \boldsymbol{\theta}^* \rangle + \langle \mathbf{m}_t(\boldsymbol{\theta}_t) - \overline{\mathbf{m}}(\boldsymbol{\theta}_t), \boldsymbol{\theta}_t - \boldsymbol{\theta}^* \rangle \\
&\quad + \langle \overline{\mathbf{m}}(\boldsymbol{\theta}_t) - \overline{\mathbf{m}}(\boldsymbol{\theta}^*), \boldsymbol{\theta}_t - \boldsymbol{\theta}^* \rangle.
\end{aligned}
\tag{6.5}
$$

Combining results from (6.4)and (6.5), we have

$$
\begin{aligned}
\|\boldsymbol{\theta}_{t+1} - \boldsymbol{\theta}^*\|_2^2 \leq{}& \|\boldsymbol{\theta}_t - \boldsymbol{\theta}^*\|_2^2 + \eta_t^2 \|\mathbf{g}_t\|_2^2 - 2\eta_t \underbrace{\langle \mathbf{g}_t - \mathbf{m}_t(\boldsymbol{\theta}_t), \boldsymbol{\theta}_t - \boldsymbol{\theta}^* \rangle}_{I_1} \\
&- 2\eta_t \underbrace{\langle \mathbf{m}_t(\boldsymbol{\theta}_t) - \overline{\mathbf{m}}(\boldsymbol{\theta}_t), \boldsymbol{\theta}_t - \boldsymbol{\theta}^* \rangle}_{I_2} - 2\eta_t \underbrace{\langle \overline{\mathbf{m}}(\boldsymbol{\theta}_t) - \overline{\mathbf{m}}(\boldsymbol{\theta}^*), \boldsymbol{\theta}_t - \boldsymbol{\theta}^* \rangle}_{I_3}.
\end{aligned}
\tag{6.6}
$$

Recall constraint set defined in (4.6). We have $\boldsymbol{\Theta} = \mathbb{B}(\boldsymbol{\theta}_0, \omega) = \{\boldsymbol{\theta} : \|\mathbf{W}_l - \mathbf{W}_l^{(0)}\|_F \leq \omega, \forall l = 1, \ldots, L\}$ and that $m$ and $\omega$ satisfy the condition in (6.3).

**Term $I_1$** is the error of the local linearization of $f(\boldsymbol{\theta})$ at $\boldsymbol{\theta}_0$. By Lemma 6.1, with probability at least $1 - 2\delta - 3L^2 \exp(-C_1 m\omega^{2/3} L)$ over the randomness of the initial point $\boldsymbol{\theta}_0$, we have

$$
|\langle \mathbf{g}_t - \mathbf{m}_t(\boldsymbol{\theta}_t), \boldsymbol{\theta}_t - \boldsymbol{\theta}^* \rangle| \leq C_2(2 + \gamma) m^{-1/6} \sqrt{\log m \log(T/\delta)}
\tag{6.7}
$$

holds uniformly for all $\boldsymbol{\theta}_t, \boldsymbol{\theta}^* \in \boldsymbol{\Theta}$, where we used the fact that $\omega = C_0 m^{-1/2} L^{-9/4}$.

**Term $I_2$** is the bias of caused by the non-i.i.d. data $(s_t, a_t, s_{t+1})$ used in the update of Algorithm 1. Conditional on the initialization, by Lemma 6.2, we have

$$
\mathbb{E}[\langle \mathbf{m}_t(\boldsymbol{\theta}_t) - \overline{\mathbf{m}}(\boldsymbol{\theta}_t), \boldsymbol{\theta}_t - \boldsymbol{\theta}^* \rangle | \boldsymbol{\theta}_0] \leq C_3(m \log(T/\delta) + m^2\omega^2) \tau^* \eta_{\max\{0, t-\tau^*\}},
\tag{6.8}
$$

where $\tau^* = \min\{t = 0, 1, 2, \ldots | \lambda\rho^t \leq \eta_T\}$ is the mixing time of the Markov chain $\{s_t, a_t\}_{t=0,1,\ldots}$.

**Term $I_3$** is the estimation error for the linear function approximation. By Lemma 6.3, we have

$$
\langle \overline{\mathbf{m}}(\boldsymbol{\theta}_t) - \overline{\mathbf{m}}(\boldsymbol{\theta}^*), \boldsymbol{\theta}_t - \boldsymbol{\theta}^* \rangle \geq \beta \mathbb{E}\big[\big(\widehat{f}(\boldsymbol{\theta}_t) - \widehat{f}(\boldsymbol{\theta}^*)\big)^2 | \boldsymbol{\theta}_0\big],
\tag{6.9}
$$

where $\beta = (1 - \alpha^{-1/2}) \in (0, 1)$ is a constant. Substituting (6.7), (6.8) and (6.9) into (6.6), we have it holds that

$$
\begin{aligned}
\|\boldsymbol{\theta}_{t+1} - \boldsymbol{\theta}^*\|_2^2 \leq{}& \|\boldsymbol{\theta}_t - \boldsymbol{\theta}^*\|_2^2 + \eta_t^2 C_4^2 (2 + \gamma)^2 m \log(T/\delta) \\
&- 2\eta_t C_2(2 + \gamma) m^{-1/6} \sqrt{\log m \log(T/\delta)} - 2\eta_t \beta \mathbb{E}\big[\big(\widehat{f}(\boldsymbol{\theta}_t) - \widehat{f}(\boldsymbol{\theta}^*)\big)^2 | \boldsymbol{\theta}_0\big] \\
&- 2\eta_t C_3(m \log(T/\delta) + m^2\omega^2) \tau^* \eta_{\max\{0, t-\tau^*\}},
\end{aligned}
\tag{6.10}
$$

with probability at least $1 - 2\delta - 3L^2 \exp(-C_1 m\omega^{2/3} L)$ over the randomness of the initial point $\boldsymbol{\theta}_0$, where we used the fact that $\|\mathbf{g}_t\|_F \leq C_4(2 + \gamma)\sqrt{m \log(T/\delta)}$ from Lemma 6.1. Rearranging the above inequality yields

$$
\begin{aligned}
\mathbb{E}\big[\big(\widehat{f}(\boldsymbol{\theta}_t) - \widehat{f}(\boldsymbol{\theta}^*)\big)^2 | \boldsymbol{\theta}_0\big] \leq{}& \frac{\|\boldsymbol{\theta}_t - \boldsymbol{\theta}^*\|_2^2 - \|\boldsymbol{\theta}_{t+1} - \boldsymbol{\theta}^*\|_2^2}{2\beta\eta_t} + \frac{C_2(2 + \gamma) m^{-1/6} \log m \log(T/\delta)}{\beta} \\
&+ \frac{C_4(2 + \gamma)^2 m \log(T/\delta) \eta_t}{\beta} + \frac{C_3 m(\log(T/\delta) + m\omega^2) \tau^* \eta_{\max\{0, t-\tau^*\}}}{\beta},
\end{aligned}
$$

with probability at least $1 - 2\delta - 3L^2 \exp(-C_1 m\omega^{2/3} L)$ over the randomness of the initial point $\boldsymbol{\theta}_0$. Recall the choices of the step sizes $\eta_0 = \ldots = \eta_T = 1/(2\beta m\sqrt{T})$ and the radius $\omega = C_0 m^{-1/2} L^{-9/4}$. Dividing the above inequality by $T$ and telescoping it from $t = 0$ to $T$ yields

$$
\begin{aligned}
\frac{1}{T} \sum_{t=0}^{T-1} \mathbb{E}\big[\big(\widehat{f}(\boldsymbol{\theta}_t) - \widehat{f}(\boldsymbol{\theta}^*)\big)^2 | \boldsymbol{\theta}_0\big] \leq{}& \frac{m\|\boldsymbol{\theta}_0 - \boldsymbol{\theta}^*\|_2^2}{\sqrt{T}} + \frac{C_2(2 + \gamma) m^{-1/6} \log m \log(T/\delta)}{\beta} \\
&+ \frac{C_4(2 + \gamma)^2 \log(T/\delta) \log T}{\beta^2 \sqrt{T}} + \frac{C_3(\log(T/\delta) + 1)\tau^* \log T}{\beta\sqrt{T}}.
\end{aligned}
$$

For $\boldsymbol{\theta}_0, \boldsymbol{\theta}^* \in \boldsymbol{\Theta}$, again by $\omega = Cm^{-1/2}L^{-9/4}$, we have $\|\boldsymbol{\theta}_0 - \boldsymbol{\theta}^*\|_2^2 \leq 1/m$. Since $\widehat{f}(\cdot) \in \mathcal{F}_{\boldsymbol{\Theta},m}$, by Lemma 6.1, it holds with probability at least $1 - 2\delta - 3L^2 \exp(-C_0 m^{2/3}L)$ over the randomness of the initial point $\boldsymbol{\theta}_0$ that

$$\frac{1}{T} \sum_{t=0}^{T-1} \mathbb{E}\big[\big(\widehat{f}(\boldsymbol{\theta}_t) - \widehat{f}(\boldsymbol{\theta}^*)\big)^2 \big| \boldsymbol{\theta}_0\big] \leq \frac{1}{\sqrt{T}} + \frac{C_1\tau^* \log(T/\delta) \log T}{\beta^2 \sqrt{T}} + \frac{C_2 \log m \log(T/\delta)}{\beta m^{1/6}},$$

where we used the fact that $\gamma < 1$. This completes the proof. $\qquad\square$

## 7 CONCLUSIONS

In this paper, we provide the first finite-time analysis of Q-learning with neural network function approximation (i.e., neural Q-learning), where the data are generated from a Markov decision process and the action-value function is approximated by a deep ReLU neural network. We prove that neural Q-learning converge to the optimal action-value function up to the approximation error with $O(1/\sqrt{T})$ rate, where $T$ is the number of iterations. Our proof technique is of independent interest and can be extended to analyze other deep reinforcement learning algorithms. One interesting future direction would be to remove the projection step in our algorithm by applying the ODE based analysis in Srikant & Ying (2019); Chen et al. (2019).

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

## A  PROOF OF THEOREM 5.6

Before we prove the global convergence of Algorithm 1, we present the following lemma that shows that near the initialization point $\boldsymbol{\theta}_0$, the neural network function $f(\boldsymbol{\theta}; \mathbf{x})$ is almost linear in $\boldsymbol{\theta}$ for all unit input vectors.

**Lemma A.1** (Theorems 5.3 and 5.4 in Cao & Gu (2019a)). *Let* $\boldsymbol{\theta}_0 = (\mathbf{W}_0^{(1)\top}, \ldots, \mathbf{W}_0^{(L)\top})^\top$ *be the initial point and* $\boldsymbol{\theta} = (\mathbf{W}^{(1)\top}, \ldots, \mathbf{W}^{(L)\top})^\top \in \mathbb{B}(\boldsymbol{\theta}_0, \omega)$ *be a point in the neighborhood of* $\boldsymbol{\theta}_0$. *If*

$$m \geq C_1 \max\{dL^2 \log(m/\delta), \omega^{-4/3} L^{-8/3} \log(m/(\omega\delta))\}, \quad \text{and} \quad \omega \leq C_2 L^{-5} (\log m)^{-3/2},$$

*then for all* $\mathbf{x} \in S^{d-1}$, *with probability at least* $1 - \delta$ *it holds that*

$$|f(\boldsymbol{\theta}; \mathbf{x}) - \widehat{f}(\boldsymbol{\theta}; \mathbf{x})| \leq \omega^{1/3} L^{8/3} \sqrt{m \log m} \sum_{l=1}^{L} \left\|\mathbf{W}^{(l)} - \mathbf{W}_0^{(l)}\right\|_2 + C_3 L^3 \sqrt{m} \sum_{l=1}^{L} \left\|\mathbf{W}^{(l)} - \mathbf{W}_0^{(l)}\right\|_2^2.$$

*Under the same conditions on* $m$ *and* $\omega$, *if* $\boldsymbol{\theta}_t \in \mathbb{B}(\boldsymbol{\theta}_0, \omega)$ *for all* $t = 1, \ldots, T$, *then with probability at least* $1 - \delta$, *we have* $|f(\boldsymbol{\theta}_t; \phi(s_t, a_t))| \leq C_4 \sqrt{\log(T/\delta)}$ *for all* $t \in [T]$.

*Proof of Theorem 5.6.* By triangle inequality, it holds that

$$Q(s, a; \boldsymbol{\theta}_T) - Q^*(s, a) \leq f(\boldsymbol{\theta}_T; s, a) - \widehat{f}(\boldsymbol{\theta}_T; s, a) + \widehat{f}(\boldsymbol{\theta}_T; s, a) - \widehat{f}(\boldsymbol{\theta}^*; s, a)$$
$$+ \widehat{f}(\boldsymbol{\theta}^*; s, a) - Q^*(s, a). \tag{A.1}$$

Recall that $\widehat{f}(\boldsymbol{\theta}^*; \cdot, \cdot)$ is the fixed point of $\Pi_{\mathcal{F}} \mathcal{T}$ and $Q^*(\cdot, \cdot)$ is the fixed point of $\mathcal{T}$. Then we have

$$\left|\widehat{f}(\boldsymbol{\theta}^*; s, a) - Q^*(s, a)\right| = \left|\widehat{f}(\boldsymbol{\theta}^*; s, a) - \Pi_{\mathcal{F}_{\boldsymbol{\Theta}, m}} Q^*(s, a) + \Pi_{\mathcal{F}_{\boldsymbol{\Theta}, m}} Q^*(s, a) - Q^*(s, a)\right|$$
$$= \left|\Pi_{\mathcal{F}_{\boldsymbol{\Theta}, m}} \mathcal{T} \widehat{f}(\boldsymbol{\theta}^*; s, a) - \Pi_{\mathcal{F}_{\boldsymbol{\Theta}, m}} \mathcal{T} Q^*(s, a) + \Pi_{\mathcal{F}_{\boldsymbol{\Theta}, m}} Q^*(s, a) - Q^*(s, a)\right|$$
$$\leq \left|\Pi_{\mathcal{F}_{\boldsymbol{\Theta}, m}} \mathcal{T} \widehat{f}(\boldsymbol{\theta}^*; s, a) - \Pi_{\mathcal{F}_{\boldsymbol{\Theta}, m}} \mathcal{T} Q^*(s, a)\right| + \left|\Pi_{\mathcal{F}_{\boldsymbol{\Theta}, m}} Q^*(s, a) - Q^*(s, a)\right|$$
$$\leq \gamma |\widehat{f}(\boldsymbol{\theta}^*; s, a) - Q^*(s, a)| + \left|\Pi_{\mathcal{F}_{\boldsymbol{\Theta}, m}} Q^*(s, a) - Q^*(s, a)\right|,$$

where the first inequality follows the triangle inequality and in the second inequality we used the fact that $\Pi_{\mathcal{F}_{\boldsymbol{\Theta}, m}} \mathcal{T}$ is $\gamma$-contractive. This further leads to

$$(1 - \gamma)|\widehat{f}(\boldsymbol{\theta}^*; s, a) - Q^*(s, a)| \leq |\Pi_{\mathcal{F}_{\boldsymbol{\Theta}, m}} Q^*(s, a) - Q^*(s, a)|.$$

To simplify the notation, we abbreviate $\mathbb{E}[\cdot|\boldsymbol{\theta}_0]$ as $\mathbb{E}[\cdot]$ in the rest of this proof. Therefore, we have

$$\mathbb{E}\big[(Q(s, a; \boldsymbol{\theta}_T) - Q^*(s, a))^2\big]$$
$$\leq 3\mathbb{E}\big[(f(\boldsymbol{\theta}_T; s, a) - \widehat{f}(\boldsymbol{\theta}_T; s, a))^2\big] + 3\mathbb{E}\big[(\widehat{f}(\boldsymbol{\theta}_T; s, a) - \widehat{f}(\boldsymbol{\theta}^*; s, a))^2\big]$$
$$+ 3\mathbb{E}\big[(\widehat{f}(\boldsymbol{\theta}^*; s, a) - Q^*(s, a))^2\big]$$
$$\leq 3\mathbb{E}\big[(f(\boldsymbol{\theta}_T; s, a) - \widehat{f}(\boldsymbol{\theta}_T; s, a))^2\big] + 3\mathbb{E}\big[(\widehat{f}(\boldsymbol{\theta}_T; s, a) - \widehat{f}(\boldsymbol{\theta}^*; s, a))^2\big]$$
$$+ 3(1 - \gamma)^{-2} \mathbb{E}\big[(\Pi_{\mathcal{F}_{\boldsymbol{\Theta}, m}} Q^*(s, a) - Q^*(s, a))^2\big].$$

By Lemma A.1 and the parameter choice that $\omega = C_1/(\sqrt{m} L^{9/4})$, we have

$$\mathbb{E}[(f(\boldsymbol{\theta}_T; s, a) - \widehat{f}(\boldsymbol{\theta}_T; s, a))^2] \leq C_2(\omega^{4/3} L^4 \sqrt{m \log m})^2 \leq C_1^{4/3} C_2 m^{-1/3} \log m$$

with probability at least $1 - \delta$. Combining the above result with Theorem 5.4, we have

$$\mathbb{E}\big[(Q(s, a; \boldsymbol{\theta}_T) - Q^*(s, a))^2\big] \leq \frac{3\mathbb{E}\big[(\Pi_{\mathcal{F}_{\boldsymbol{\Theta}, m}} Q^*(s, a) - Q^*(s, a))^2\big]}{(1 - \gamma)^2} + \frac{1}{\sqrt{T}}$$
$$+ \frac{C_2 \tau^* \log(T/\delta) \log T}{\beta^2 \sqrt{T}} + \frac{C_3 \log(T/\delta) \log m}{\beta m^{1/6}},$$

with probability at least $1 - 3\delta - L^2 \exp(-C_6 m^{2/3} L)$, which completes the proof. $\square$

## B  PROOF OF SUPPORTING LEMMAS

### B.1  PROOF OF LEMMA 6.1

Before we prove the error bound for the local linearization, we first present some useful lemmas from recent studies of overparameterized deep neural networks. Note that in the following lemmas, $\{C_i\}_{i=1,\dots}$ are universal constants that are independent of problem parameters such as $d, \boldsymbol{\theta}, m, L$ and their values can be different in different contexts. The first lemma states the uniform upper bound for the gradient of the deep neural network. Note that by definition, our parameter $\boldsymbol{\theta}$ is a long vector containing the concatenation of the vectorization of all the weight matrices. Correspondingly, the gradient $\nabla_{\boldsymbol{\theta}} f(\boldsymbol{\theta}; \mathbf{x})$ is also a long vector.

**Lemma B.1** (Lemma B.3 in Cao & Gu (2019b)). Let $\boldsymbol{\theta} \in \mathbb{B}(\boldsymbol{\theta}_0, \omega)$ with the radius satisfying $C_1 d^{3/2} L^{-1} m^{-3/2} \le \omega \le C_2 L^{-6} (\log m)^{-3/2}$. Then for all unit vectors in $\mathbb{R}^d$, i.e., $\mathbf{x} \in S^{d-1}$, the gradient of the neural network $f$ defined in (4.2) is bounded as $\|\nabla_{\boldsymbol{\theta}} f(\boldsymbol{\theta}; \mathbf{x})\|_2 \le C_3 \sqrt{m}$ with probability at least $1 - L^2 \exp(-C_4 m \omega^{2/3} L)$.

The second lemma provides the perturbation bound for the gradient of the neural network function. Note that the original theorem holds for any fixed $d$ dimensional unit vector $\mathbf{x}$. However, due to the choice of $\omega$ and its dependency on $m$ and $d$, it is easy to modify the results to hold for all $\mathbf{x} \in S^{d-1}$.

**Lemma B.2** (Theorem 5 in Allen-Zhu et al. (2019b)). Let $\boldsymbol{\theta} \in \mathbb{B}(\boldsymbol{\theta}_0, \omega)$ with the radius satisfying

$$C_1 d^{3/2} L^{-3/2} m^{-3/2} (\log m)^{-3/2} \le \omega \le C_2 L^{-9/2} (\log m)^{-3}.$$

Then for all $\mathbf{x} \in S^{d-1}$, with probability at least $1 - \exp(-C_3 m \omega^{2/3} L)$ over the randomness of $\boldsymbol{\theta}_0$, it holds that

$$\|\nabla_{\boldsymbol{\theta}} f(\boldsymbol{\theta}; \mathbf{x}) - \nabla_{\boldsymbol{\theta}} f(\boldsymbol{\theta}_0; \mathbf{x})\|_2 \le C_4 \omega^{1/3} L^3 \sqrt{\log m} \|\nabla_{\boldsymbol{\theta}} f(\boldsymbol{\theta}_0; \mathbf{x})\|_2.$$

Now we are ready to bound the linearization error.

*Proof of Lemma 6.1.* Recall the definition of $\mathbf{g}_t(\boldsymbol{\theta}_t)$ and $\mathbf{m}_t(\boldsymbol{\theta}_t)$ in (4.5) and (6.2) respectively. We have

$$\|\mathbf{g}_t(\boldsymbol{\theta}_t) - \mathbf{m}_t(\boldsymbol{\theta}_t)\|_2 = \left\| \nabla_{\boldsymbol{\theta}} f(\boldsymbol{\theta}_t; s_t, a_t) \Delta(s_t, a_t, s_{t+1}; \boldsymbol{\theta}_t) - \nabla_{\boldsymbol{\theta}} \widehat{f}(\boldsymbol{\theta}_t; s_t, a_t) \widehat{\Delta}(s_t, a_t, s_{t+1}; \boldsymbol{\theta}_t) \right\|_2$$

$$\le \left\| (\nabla_{\boldsymbol{\theta}} f(\boldsymbol{\theta}_t; s_t, a_t) - \nabla_{\boldsymbol{\theta}} \widehat{f}(\boldsymbol{\theta}_t; s_t, a_t)) \Delta(s_t, a_t, s_{t+1}; \boldsymbol{\theta}_t) \right\|_2$$

$$+ \left\| \nabla_{\boldsymbol{\theta}} \widehat{f}(\boldsymbol{\theta}_t; s_t, a_t) \big( \Delta(s_t, a_t, s_{t+1}; \boldsymbol{\theta}_t) - \widehat{\Delta}(s_t, a_t, s_{t+1}; \boldsymbol{\theta}_t) \big) \right\|_2. \quad (B.1)$$

Since $\widehat{f}(\boldsymbol{\theta}) \in \mathcal{F}_{\boldsymbol{\Theta}, m}$, we have $\widehat{f}(\boldsymbol{\theta}) = f(\boldsymbol{\theta}_0) + \langle \nabla_{\boldsymbol{\theta}} f(\boldsymbol{\theta}_0), \boldsymbol{\theta} - \boldsymbol{\theta}_0 \rangle$ and $\nabla_{\boldsymbol{\theta}} \widehat{f}(\boldsymbol{\theta}) = \nabla_{\boldsymbol{\theta}} f(\boldsymbol{\theta}_0)$. Then with probability at least $1 - 2L^2 \exp(-C_1 m \omega^{2/3} L)$, we have

$$\left\| (\nabla_{\boldsymbol{\theta}} f(\boldsymbol{\theta}_t; s_t, a_t) - \nabla_{\boldsymbol{\theta}} \widehat{f}(\boldsymbol{\theta}_t; s_t, a_t)) \Delta(s_t, a_t, s_{t+1}; \boldsymbol{\theta}_t) \right\|_2$$

$$= |\Delta(s_t, a_t, s_{t+1}; \boldsymbol{\theta}_t)| \cdot \left\| (\nabla_{\boldsymbol{\theta}} f(\boldsymbol{\theta}_t; s_t, a_t) - \nabla_{\boldsymbol{\theta}} f(\boldsymbol{\theta}_0; s_t, a_t)) \right\|_2$$

$$\le C_2 \omega^{1/3} L^3 \sqrt{m \log m} |\Delta(s_t, a_t, s_{t+1}; \boldsymbol{\theta}_t)|,$$

where the inequality comes from Lemmas B.1 and B.2. By Lemma A.1, with probability at least $1 - \delta$, it holds that

$$|\Delta(s_t, a_t, s_{t+1}; \boldsymbol{\theta}_t)| = \left| f(\boldsymbol{\theta}_t; s_t, a_t) - r_t - \gamma \max_{b \in \mathcal{A}} f(\boldsymbol{\theta}_t; s_{t+1}, b) \right| \le (2 + \gamma) C_3 \sqrt{\log(T/\delta)},$$

which further implies that with probability at least $1 - \delta - 2L^2 \exp(-C_1 m \omega^{2/3} L)$, we have

$$\left\| (\nabla_{\boldsymbol{\theta}} f(\boldsymbol{\theta}_t; s_t, a_t) - \nabla_{\boldsymbol{\theta}} \widehat{f}(\boldsymbol{\theta}_t; s_t, a_t)) \Delta(s_t, a_t, s_{t+1}; \boldsymbol{\theta}_t) \right\|_2$$

$$\le (2 + \gamma) C_2 C_3 \omega^{1/3} L^3 \sqrt{m \log m \log(T/\delta)}.$$

For the second term in (B.1), we have

$$\left\| \nabla_{\boldsymbol{\theta}} \widehat{f}(\boldsymbol{\theta}_t; s_t, a_t) \big( \Delta(s_t, a_t, s_{t+1}; \boldsymbol{\theta}_t) - \widehat{\Delta}(s_t, a_t, s_{t+1}; \boldsymbol{\theta}_t) \big) \right\|_2$$

$$\leq \left\| \nabla_{\boldsymbol{\theta}} \widehat{f}(\boldsymbol{\theta}_t; s_t, a_t) \big( f(\boldsymbol{\theta}_t; s_t, a_t) - \widehat{f}(\boldsymbol{\theta}_t; s_t, a_t) \big) \right\|_2$$
$$+ \left\| \nabla_{\boldsymbol{\theta}} \widehat{f}(\boldsymbol{\theta}_t; s_t, a_t) \Big( \max_{b \in \mathcal{A}} f(\boldsymbol{\theta}_t; s_{t+1}, b) - \max_{b \in \mathcal{A}} \widehat{f}(\boldsymbol{\theta}_t; s_{t+1}, b) \Big) \right\|_2$$
$$\leq \left\| \nabla_{\boldsymbol{\theta}} \widehat{f}(\boldsymbol{\theta}_t; s_t, a_t) \right\|_2 \cdot \left| f(\boldsymbol{\theta}_t; s_t, a_t) - \widehat{f}(\boldsymbol{\theta}_t; s_t, a_t) \right|$$
$$+ \|\nabla_{\boldsymbol{\theta}} \widehat{f}(\boldsymbol{\theta}_t; s_t, a_t)\|_2 \max_{b \in \mathcal{A}} \left| f(\boldsymbol{\theta}_t; s_{t+1}, b) - \widehat{f}(\boldsymbol{\theta}; s_{t+1}, b) \right|. \tag{B.2}$$

By Lemma A.1, with probability at least $1 - \delta$ we have

$$|f(\boldsymbol{\theta}_t; s_t, a_t) - \widehat{f}(\boldsymbol{\theta}_t; s_t, a_t)| \leq \omega^{4/3} L^{11/3} \sqrt{m \log m} + C_4 \omega^2 L^4 \sqrt{m},$$

for all $(s_t, a_t) \in \mathcal{S} \times \mathcal{A}$ such that $\|\phi(s_t, a_t)\|_2 = 1$. Substituting the above result into (B.2) and applying the gradient bound in Lemma B.1, we obtain with probability at least $1 - \delta - L^2 \exp(-C_1 m \omega^{2/3} L)$ that

$$\left\| \nabla_{\boldsymbol{\theta}} \widehat{f}(\boldsymbol{\theta}_t; s_t, a_t) \big( \Delta(s_t, a_t, s_{t+1}; \boldsymbol{\theta}_t) - \widehat{\Delta}(s_t, a_t, s_{t+1}; \boldsymbol{\theta}_t) \big) \right\|_2$$
$$\leq C_5 \omega^{4/3} L^{11/3} m \sqrt{\log m} + C_6 \omega^2 L^4 m.$$

Note that the above results require that the choice of $\omega$ should satisfy all the constraints in Lemmas B.1, A.1 and B.2, of which the intersection is

$$C_7 d^{3/2} L^{-1} m^{-3/4} \leq \omega \leq C_8 L^{-6} (\log m)^{-3}.$$

Therefore, the error of the local linearization of $\mathbf{g}_t(\boldsymbol{\theta}_t)$ can be upper bounded by

$$|\langle \mathbf{g}(\boldsymbol{\theta}_t) - \mathbf{m}(\boldsymbol{\theta}_t), \boldsymbol{\theta}_t - \boldsymbol{\theta}^* \rangle| \leq (2 + \gamma) C_2 C_3 \omega^{1/3} L^3 \sqrt{m \log m \log(T/\delta)} \|\boldsymbol{\theta}_t - \boldsymbol{\theta}^*\|_2$$
$$+ \big( C_5 \omega^{4/3} L^{11/3} m \sqrt{\log m} + C_6 \omega^2 L^4 m \big) \|\boldsymbol{\theta}_t - \boldsymbol{\theta}^*\|_2,$$

which holds with probability at least $1 - 2\delta - 3L^2 \exp(-C_1 m \omega^{2/3} L)$ over the randomness of the initial point. For the upper bound of the norm of $\mathbf{g}_t$, by Lemmas B.1 and A.1, we have

$$\|\mathbf{g}_t\|_2 = \left\| \nabla_{\boldsymbol{\theta}} f(\boldsymbol{\theta}_t; s_t, a_t) \Big( f(\boldsymbol{\theta}_t; s_t, a_t) - r_t - \gamma \max_{b \in \mathcal{A}} f(\boldsymbol{\theta}_t; s_{t+1}, b) \Big) \right\|_2$$
$$\leq (2 + \gamma) C_9 \sqrt{m \log(T/\delta)}$$

holds with probability at least $1 - \delta - L^2 \exp(-C_1 m \omega^{2/3} L)$. □

## B.2 Proof of Lemma 6.2

Let us define $\zeta_t(\boldsymbol{\theta}) = \langle \mathbf{m}_t(\boldsymbol{\theta}) - \overline{\mathbf{m}}(\boldsymbol{\theta}), \boldsymbol{\theta} - \boldsymbol{\theta}^* \rangle$, which characterizes the bias of the data. Different from the similar quantity $\zeta_t$ in Bhandari et al. (2018), our definition is based on the local linearization of $f$, which is essential to the analysis in our proof. It is easy to verify that $\mathbb{E}[\mathbf{m}_t(\boldsymbol{\theta})] = \overline{\mathbf{m}}(\boldsymbol{\theta})$ for any fixed and deterministic $\boldsymbol{\theta}$. However, it should be noted that $\mathbb{E}[\mathbf{m}_t(\boldsymbol{\theta}_t)|\boldsymbol{\theta}_t = \boldsymbol{\theta}] \neq \overline{\mathbf{m}}(\boldsymbol{\theta})$ because $\boldsymbol{\theta}_t$ depends on all historical states and actions $\{s_t, a_t, s_{t-1}, a_{t-1}, \ldots\}$ and $\mathbf{m}_t(\cdot)$ depends on the current observation $\{s_t, a_t, s_{t+1}\}$ and thus also depends on $\{s_{t-1}, a_{t-1}, s_{t-2}, a_{t-2}, \ldots\}$. Therefore, we need a careful analysis of Markov chains to decouple the dependency between $\boldsymbol{\theta}_t$ and $\mathbf{m}_t(\cdot)$.

The following lemma uses data processing inequality to provide an information theoretic control of coupling.

**Lemma B.3** (Control of coupling, (Bhandari et al., 2018)). Consider two random variables $X$ and $Y$ that form the following Markov chain:

$$X \to s_t \to s_{t+\tau} \to Y,$$

where $t \in \{0, 1, 2, \ldots\}$ and $\tau > 0$. Suppose Assumption 5.2 holds. Let $X'$ and $Y'$ be independent copies drawn from the marginal distributions of $X$ and $Y$ respectively, i.e., $\mathbb{P}(X' = \cdot, Y' = \cdot) = \mathbb{P}(X = \cdot) \otimes \mathbb{P}(Y = \cdot)$. Then for any bounded function $h : \mathcal{S} \times \mathcal{S} \to \mathbb{R}$, it holds that

$$|\mathbb{E}[h(X, Y)] - \mathbb{E}[h(X', Y')]| \leq 2 \sup_{s, s'} |h(s, s')| \lambda \rho^\tau.$$

*Proof of Lemma 6.2.* The proof of this lemma is adapted from Bhandari et al. (2018), where the result was originally proved for linear function approximation of temporal difference learning. We first show that $\zeta_t(\boldsymbol{\theta})$ is Lipschitz. For any $\boldsymbol{\theta}, \boldsymbol{\theta}' \in \mathbb{B}(\boldsymbol{\theta}_0, \omega)$, we have

$$\zeta_t(\boldsymbol{\theta}) - \zeta_t(\boldsymbol{\theta}') = \langle \mathbf{m}_t(\boldsymbol{\theta}) - \overline{\mathbf{m}}(\boldsymbol{\theta}), \boldsymbol{\theta} - \boldsymbol{\theta}^* \rangle - \langle \mathbf{m}_t(\boldsymbol{\theta}') - \overline{\mathbf{m}}(\boldsymbol{\theta}'), \boldsymbol{\theta}' - \boldsymbol{\theta}^* \rangle$$
$$= \langle \mathbf{m}_t(\boldsymbol{\theta}) - \overline{\mathbf{m}}(\boldsymbol{\theta}) - (\mathbf{m}_t(\boldsymbol{\theta}') - \overline{\mathbf{m}}(\boldsymbol{\theta}')), \boldsymbol{\theta} - \boldsymbol{\theta}^* \rangle$$
$$+ \langle \mathbf{m}_t(\boldsymbol{\theta}') - \overline{\mathbf{m}}(\boldsymbol{\theta}'), \boldsymbol{\theta} - \boldsymbol{\theta}' \rangle,$$

which directly implies

$$|\zeta_t(\boldsymbol{\theta}) - \zeta_t(\boldsymbol{\theta}')| \le \|\mathbf{m}_t(\boldsymbol{\theta}) - \mathbf{m}_t(\boldsymbol{\theta}')\|_2 \cdot \|\boldsymbol{\theta} - \boldsymbol{\theta}^*\|_2 + \|\overline{\mathbf{m}}(\boldsymbol{\theta}) - \overline{\mathbf{m}}(\boldsymbol{\theta}')\|_2 \cdot \|\boldsymbol{\theta} - \boldsymbol{\theta}^*\|_2$$
$$+ \|\mathbf{m}_t(\boldsymbol{\theta}') - \overline{\mathbf{m}}(\boldsymbol{\theta}')\|_2 \cdot \|\boldsymbol{\theta} - \boldsymbol{\theta}'\|_2.$$

By the definition of $\mathbf{m}_t$, we have

$$\|\mathbf{m}_t(\boldsymbol{\theta}) - \mathbf{m}_t(\boldsymbol{\theta}')\|_2$$
$$= \left\| \nabla_{\boldsymbol{\theta}} f(\boldsymbol{\theta}_0) \Big( \big( f(\boldsymbol{\theta}; s, a) - f(\boldsymbol{\theta}'; s, a) \big) - \gamma \Big( \max_{b \in \mathcal{A}} f(\boldsymbol{\theta}; s', b) - \max_{b \in \mathcal{A}} f(\boldsymbol{\theta}'; s', b) \Big) \Big) \right\|_2$$
$$\le (1 + \gamma) C_3^2 m \|\boldsymbol{\theta} - \boldsymbol{\theta}'\|_2,$$

which holds with probability at least $1 - L^2 \exp(-C_4 m \omega^{2/3} L)$, where we used the fact that the neural network function is Lipschitz with parameter $C_3 \sqrt{m}$ by Lemma B.1. Similar bound can also be established for $\|\overline{\mathbf{m}}_t(\boldsymbol{\theta}) - \overline{\mathbf{m}}_t(\boldsymbol{\theta}')\|$ in the same way. Note that for $\boldsymbol{\theta} \in \mathbb{B}(\boldsymbol{\theta}_0, \omega)$ with $\omega$ and $m$ satisfying the conditions in Lemma 6.1, we have by the definition in (6.2) that

$$\|\mathbf{m}_t(\boldsymbol{\theta})\|_2 \le \Big( |\widehat{f}(\boldsymbol{\theta}; s, a)| + r(s, a) + \gamma \big| \max_b \widehat{f}(\boldsymbol{\theta}; s', b) \big| \Big) \|\nabla_{\boldsymbol{\theta}} \widehat{f}(\boldsymbol{\theta})\|_2$$
$$\le 2(2 + \gamma)(|f(\boldsymbol{\theta}_0)| + \|\nabla_{\boldsymbol{\theta}} f(\boldsymbol{\theta}_0)\|_2 \cdot \|\boldsymbol{\theta} - \boldsymbol{\theta}_0\|_2) \|\nabla_{\boldsymbol{\theta}} f(\boldsymbol{\theta}_0)\|_2$$
$$\le 2(2 + \gamma) C_3 (C_8 \sqrt{m} \sqrt{\log(T/\delta)} + C_3 m \omega).$$

The same bound can be established for $\|\overline{\mathbf{m}}_t\|$ in a similar way. Therefore, we have $|\zeta_t(\boldsymbol{\theta}) - \zeta_t(\boldsymbol{\theta}')| \le \ell_{m,L} \|\boldsymbol{\theta} - \boldsymbol{\theta}'\|_2$, where $\ell_{m,L}$ is defined as

$$\ell_{m,L} = 2(1 + \gamma) C_3^2 m \omega + 2(2 + \gamma) C_3 (C_8 \sqrt{m} \sqrt{\log(T/\delta)} + C_3 m \omega).$$

Applying the above inequality recursively, for all $\tau = 0, \ldots, t$, we have

$$\zeta_t(\boldsymbol{\theta}_t) \le \zeta_t(\boldsymbol{\theta}_{t-\tau}) + \ell_{m,L} \sum_{i=t-\tau}^{t-1} \|\boldsymbol{\theta}_{i+1} - \boldsymbol{\theta}_i\|_2$$

$$\le \zeta_t(\boldsymbol{\theta}_{t-\tau}) + 2(2 + \gamma) C_3 (C_8 \sqrt{m} \sqrt{\log(T/\delta)} + C_3 m \omega) \ell_{m,L} \sum_{i=t-\tau}^{t-1} \eta_i. \quad \text{(B.3)}$$

Next, we need to bound $\zeta_t(\boldsymbol{\theta}_{t-\tau})$. Define the observed tuple $O_t = (s_t, a_t, s_{t+1})$ as the collection of the current state and action and the next state. Note that $\boldsymbol{\theta}_{t-\tau} \to s_{t-\tau} \to s_t \to O_t$ forms a Markov chain induced by the target policy $\pi$. Recall that $\mathbf{m}_t(\cdot)$ depends on the observation $O_t$. Let's rewrite $\mathbf{m}(\boldsymbol{\theta}, O_t) = \mathbf{m}_t(\boldsymbol{\theta})$. Similarly, we can rewrite $\zeta_t(\boldsymbol{\theta})$ as $\zeta(\boldsymbol{\theta}, O_t)$. Let $\boldsymbol{\theta}_t'$ and $O_t'$ be independently drawn from the marginal distributions of $\boldsymbol{\theta}_t$ and $O_t$ respectively. Applying Lemma B.3 yields

$$\mathbb{E}[\zeta(\boldsymbol{\theta}_{t-\tau}, O_t)] - \mathbb{E}[\zeta(\boldsymbol{\theta}_{t-\tau}', O_t')] \le 2 \sup_{\boldsymbol{\theta}, O} |\zeta(\boldsymbol{\theta}, O)| \lambda \rho^\tau,$$

where we used the uniform mixing result in Assumption 5.2. By definition $\boldsymbol{\theta}_{t-\tau}'$ and $O_t'$ are independent, which implies $\mathbb{E}[\mathbf{m}(\boldsymbol{\theta}_t', O_t')|\boldsymbol{\theta}_t'] = \overline{\mathbf{m}}(\boldsymbol{\theta}_t')$ and

$$\mathbb{E}[\zeta(\boldsymbol{\theta}_{t-\tau}', O_t')] = \mathbb{E}[\mathbb{E}[\langle \mathbf{m}(\boldsymbol{\theta}_t', O_t') - \overline{\mathbf{m}}(\boldsymbol{\theta}_t'), \boldsymbol{\theta}_t' - \boldsymbol{\theta}^* \rangle]|\boldsymbol{\theta}_t'] = 0.$$

Therefore, for any $\tau = 0, \ldots, t$, we have

$$\mathbb{E}[\zeta_t(\boldsymbol{\theta}_t)] \le \mathbb{E}\zeta_t(\boldsymbol{\theta}_{t-\tau}) + 2(2 + \gamma) C_3 (C_8 \sqrt{m} \sqrt{\log(T/\delta)} + C_3 m \omega) \ell_{m,L} \sum_{i=t-\tau}^{t-1} \eta_i$$

$$\leq 2\sup \lambda \rho^\tau + 2(2+\gamma)C_3(C_8\sqrt{m}\sqrt{\log(T/\delta)} + C_3 m\omega)\ell_{m,L}\tau\eta_{t-\tau}. \tag{B.4}$$

Define $\tau^*$ as the mixing time of the Markov chain that satisfies

$$\tau^* = \min\{t = 0, 1, 2, \ldots | \lambda\rho^t \leq \eta_T\}.$$

When $t \leq \tau^*$, we choose $\tau = t$ in (B.4) and obtain

$$\mathbb{E}[\zeta_t(\boldsymbol{\theta}_t)] \leq \mathbb{E}[\zeta_t(\boldsymbol{\theta}_0)] + 2(2+\gamma)C_3(C_8\sqrt{m}\sqrt{\log(T/\delta)} + C_3 m\omega)\ell_{m,L}\tau^*\eta_0$$
$$= 2(2+\gamma)C_3(C_8\sqrt{m}\sqrt{\log(T/\delta)} + C_3 m\omega)\ell_{m,L}\tau^*\eta_0,$$

where we used the fact that the initial point $\boldsymbol{\theta}_0$ is independent of $\{s_t, a_t, s_{t-1}, a_{t-1}, \ldots, s_0, a_0\}$ and thus independent of $\zeta_t(\cdot)$. When $t > \tau^*$, we can choose $\tau = \tau^*$ in (B.4) and obtain

$$\mathbb{E}[\zeta_t(\boldsymbol{\theta}_t)] \leq 2\eta_T + 2(2+\gamma)C_3(C_8\sqrt{m}\sqrt{\log(T/\delta)} + C_3 m\omega)\ell_{m,L}\tau^*\eta_{t-\tau^*}$$
$$\leq \widetilde{C}(m\log(T/\delta) + m^2\omega^2)\tau^*\eta_{t-\tau^*},$$

where $\widetilde{C} > 0$ is a universal constant, which completes the proof. $\qquad\square$

### B.3 PROOF OF LEMMA 6.3

*Proof of Lemma 6.3.* To simplify the notation, we use $\mathbb{E}_\pi$ to denote $\mathbb{E}_{\mu,\pi,\mathcal{P}}$, namely, the expectation over $s \in \mu, a \sim \pi(\cdot|s)$ and $s' \sim \mathcal{P}(\cdot|s,a)$, in the rest of the proof. By the definition of $\overline{\mathbf{m}}$ in (6.2), we have

$$\langle \overline{\mathbf{m}}(\boldsymbol{\theta}) - \overline{\mathbf{m}}(\boldsymbol{\theta}^*), \boldsymbol{\theta} - \boldsymbol{\theta}^* \rangle$$
$$= \mathbb{E}_\pi\big[\big(\widehat{\Delta}(s,a,s';\boldsymbol{\theta}) - \widehat{\Delta}(s,a,s';\boldsymbol{\theta}^*)\big)\langle \nabla_{\boldsymbol{\theta}} f(\boldsymbol{\theta}_0; s,a), \boldsymbol{\theta} - \boldsymbol{\theta}^* \rangle\big]$$
$$= \mathbb{E}_\pi\big[\big(\widehat{f}(\boldsymbol{\theta}; s,a) - \widehat{f}(\boldsymbol{\theta}^*; s,a)\big)\langle \nabla_{\boldsymbol{\theta}} f(\boldsymbol{\theta}_0; s,a), \boldsymbol{\theta} - \boldsymbol{\theta}^* \rangle\big]$$
$$- \gamma\mathbb{E}_\pi\Big[\Big(\max_{b\in\mathcal{A}} \widehat{f}(\boldsymbol{\theta}; s',b) - \max_{b\in\mathcal{A}} \widehat{f}(\boldsymbol{\theta}^*; s',b)\Big)\langle \nabla_{\boldsymbol{\theta}} f(\boldsymbol{\theta}_0; s,a), \boldsymbol{\theta} - \boldsymbol{\theta}^* \rangle\Big],$$

where in the first equation we used the fact that $\nabla_{\boldsymbol{\theta}}\widehat{f}(\boldsymbol{\theta}) = \nabla_{\boldsymbol{\theta}} f(\boldsymbol{\theta}_0)$ for all $\boldsymbol{\theta} \in \Theta$ and $\widehat{f} \in \mathcal{F}_{\Theta,m}$. Further by the property of the local linearization of $f$ at $\boldsymbol{\theta}_0$, we have

$$\widehat{f}(\boldsymbol{\theta}; s,a) - \widehat{f}(\boldsymbol{\theta}^*; s,a) = \langle \nabla_{\boldsymbol{\theta}} f(\boldsymbol{\theta}_0; s,a), \boldsymbol{\theta} - \boldsymbol{\theta}^* \rangle, \tag{B.5}$$

which further implies

$$\mathbb{E}\big[\big(\widehat{f}(\boldsymbol{\theta}; s,a) - \widehat{f}(\boldsymbol{\theta}^*; s,a)\big)\langle \nabla_{\boldsymbol{\theta}} f(\boldsymbol{\theta}_0; s,a), \boldsymbol{\theta} - \boldsymbol{\theta}^* \rangle|\boldsymbol{\theta}_0\big]$$
$$= (\boldsymbol{\theta} - \boldsymbol{\theta}^*)^\top \mathbb{E}\big[\nabla_{\boldsymbol{\theta}} f(\boldsymbol{\theta}_0; s,a)\nabla_{\boldsymbol{\theta}} f(\boldsymbol{\theta}_0; s,a)^\top|\boldsymbol{\theta}_0\big](\boldsymbol{\theta} - \boldsymbol{\theta}^*)$$
$$= m\|\boldsymbol{\theta} - \boldsymbol{\theta}^*\|_{\Sigma_\pi}^2.$$

where $\Sigma_\pi$ is defined in Assumption 5.3. For the other term, we define $b_{\max}(\boldsymbol{\theta}) = \arg\max_{b\in\mathcal{A}} \widehat{f}(\boldsymbol{\theta}; s',b)$ and $b_{\max}(\boldsymbol{\theta}^*) = \arg\max_{b\in\mathcal{A}} \widehat{f}(\boldsymbol{\theta}^*; s',b)$. Then we have

$$\mathbb{E}_\pi\Big[\Big(\max_{b\in\mathcal{A}} \widehat{f}(\boldsymbol{\theta}; s',b) - \max_{b\in\mathcal{A}} \widehat{f}(\boldsymbol{\theta}^*; s',b)\Big)\langle \nabla_{\boldsymbol{\theta}} f(\boldsymbol{\theta}_0; s,a), \boldsymbol{\theta} - \boldsymbol{\theta}^* \rangle\Big]$$
$$= \mathbb{E}_\pi\big[\big(\widehat{f}(\boldsymbol{\theta}; s', b_{\max}) - \widehat{f}(\boldsymbol{\theta}^*; s', b_{\max}^*)\big)\langle \nabla_{\boldsymbol{\theta}} f(\boldsymbol{\theta}_0; s,a), \boldsymbol{\theta} - \boldsymbol{\theta}^* \rangle\big]. \tag{B.6}$$

For all $(s,a,s')$, when $\langle \nabla_{\boldsymbol{\theta}} f(\boldsymbol{\theta}_0; s,a), \boldsymbol{\theta} - \boldsymbol{\theta}^* \rangle \geq 0$, (B.6) can be upper bounded by

$$\big(\widehat{f}(\boldsymbol{\theta}; s', b_{\max}) - \widehat{f}(\boldsymbol{\theta}^*; s', b_{\max}^*)\big)\langle \nabla_{\boldsymbol{\theta}} f(\boldsymbol{\theta}_0; s,a), \boldsymbol{\theta} - \boldsymbol{\theta}^* \rangle$$
$$= \big(\widehat{f}(\boldsymbol{\theta}; s', b_{\max}) - \widehat{f}(\boldsymbol{\theta}^*; s', b_{\max}) + \widehat{f}(\boldsymbol{\theta}^*; s', b_{\max}) - \widehat{f}(\boldsymbol{\theta}^*; s', b_{\max}^*)\big)\langle \nabla_{\boldsymbol{\theta}} f(\boldsymbol{\theta}_0; s,a), \boldsymbol{\theta} - \boldsymbol{\theta}^* \rangle$$
$$\leq \big(\widehat{f}(\boldsymbol{\theta}; s', b_{\max}) - \widehat{f}(\boldsymbol{\theta}^*; s', b_{\max})\big)\langle \nabla_{\boldsymbol{\theta}} f(\boldsymbol{\theta}_0; s,a), \boldsymbol{\theta} - \boldsymbol{\theta}^* \rangle$$
$$= (\boldsymbol{\theta} - \boldsymbol{\theta}^*)^\top \nabla_{\boldsymbol{\theta}} f(\boldsymbol{\theta}_0; s', b_{\max})\nabla_{\boldsymbol{\theta}} f(\boldsymbol{\theta}_0; s,a)^\top(\boldsymbol{\theta} - \boldsymbol{\theta}^*)$$
$$\leq |(\boldsymbol{\theta} - \boldsymbol{\theta}^*)^\top \nabla_{\boldsymbol{\theta}} f(\boldsymbol{\theta}_0; s', b_{\max})| \cdot |\nabla_{\boldsymbol{\theta}} f(\boldsymbol{\theta}_0; s,a)^\top(\boldsymbol{\theta} - \boldsymbol{\theta}^*)|,$$

where the inequality comes from the optimality of $b_{\max}^*$ and the last equality follows the fact that $\widehat{f}(\boldsymbol{\theta}; \cdot, \cdot)$ is linear. When $\langle \nabla_{\boldsymbol{\theta}} f(\boldsymbol{\theta}_0; s, a), \boldsymbol{\theta} - \boldsymbol{\theta}^* \rangle < 0$, using the same argument, we can upper bound (B.6) as follows

$$
\begin{aligned}
&\big(\widehat{f}(\boldsymbol{\theta}; s', b_{\max}) - \widehat{f}(\boldsymbol{\theta}^*; s', b_{\max}^*)\big)\langle \nabla_{\boldsymbol{\theta}} f(\boldsymbol{\theta}_0; s, a), \boldsymbol{\theta} - \boldsymbol{\theta}^* \rangle \\
&= \big(\widehat{f}(\boldsymbol{\theta}; s', b_{\max}) - \widehat{f}(\boldsymbol{\theta}; s', b_{\max}^*) + \widehat{f}(\boldsymbol{\theta}; s', b_{\max}^*) - \widehat{f}(\boldsymbol{\theta}^*; s', b_{\max}^*)\big)\langle \nabla_{\boldsymbol{\theta}} f(\boldsymbol{\theta}_0; s, a), \boldsymbol{\theta} - \boldsymbol{\theta}^* \rangle \\
&\leq \big(\widehat{f}(\boldsymbol{\theta}; s', b_{\max}^*) - \widehat{f}(\boldsymbol{\theta}^*; s', b_{\max}^*)\big)\langle \nabla_{\boldsymbol{\theta}} f(\boldsymbol{\theta}_0; s, a), \boldsymbol{\theta} - \boldsymbol{\theta}^* \rangle \\
&\leq |(\boldsymbol{\theta} - \boldsymbol{\theta}^*)^\top \nabla_{\boldsymbol{\theta}} f(\boldsymbol{\theta}_0; s', b_{\max}^*)| \cdot |\nabla_{\boldsymbol{\theta}} f(\boldsymbol{\theta}_0; s, a)^\top (\boldsymbol{\theta} - \boldsymbol{\theta}^*)|.
\end{aligned}
$$

Combining the above result, we have for all tuples $(s, a, s')$ it holds that

$$
\begin{aligned}
&\big(\widehat{f}(\boldsymbol{\theta}; s', b_{\max}) - \widehat{f}(\boldsymbol{\theta}^*; s', b_{\max}^*)\big)\langle \nabla_{\boldsymbol{\theta}} f(\boldsymbol{\theta}_0; s, a), \boldsymbol{\theta} - \boldsymbol{\theta}^* \rangle \\
&\leq |(\boldsymbol{\theta} - \boldsymbol{\theta}^*)^\top \nabla_{\boldsymbol{\theta}} f(\boldsymbol{\theta}_0; s', b_{\max})| \cdot |\nabla_{\boldsymbol{\theta}} f(\boldsymbol{\theta}_0; s, a)^\top (\boldsymbol{\theta} - \boldsymbol{\theta}^*)|\mathbb{1}^+ \\
&\quad + |(\boldsymbol{\theta} - \boldsymbol{\theta}^*)^\top \nabla_{\boldsymbol{\theta}} f(\boldsymbol{\theta}_0; s', b_{\max}^*)| \cdot |\nabla_{\boldsymbol{\theta}} f(\boldsymbol{\theta}_0; s, a)^\top (\boldsymbol{\theta} - \boldsymbol{\theta}^*)|\mathbb{1}^-,
\end{aligned}
$$

where we denote $\mathbb{1}^+ = \mathbb{1}\{\langle \nabla_{\boldsymbol{\theta}} f(\boldsymbol{\theta}_0; s, a), \boldsymbol{\theta} - \boldsymbol{\theta}^* \rangle \geq 0\}$ and $\mathbb{1}^- = \mathbb{1}\{\langle \nabla_{\boldsymbol{\theta}} f(\boldsymbol{\theta}_0; s, a), \boldsymbol{\theta} - \boldsymbol{\theta}^* \rangle < 0\}$. Taking expectation over the above inequality and applying Cauchy-Schwarz inequality, we have

$$
\begin{aligned}
&\mathbb{E}_{\mu, \pi, \mathcal{P}}\big[\big(\widehat{f}(\boldsymbol{\theta}; s', b_{\max}) - \widehat{f}(\boldsymbol{\theta}^*; s', b_{\max}^*)\big)\langle \nabla_{\boldsymbol{\theta}} f(\boldsymbol{\theta}_0; s, a), \boldsymbol{\theta} - \boldsymbol{\theta}^* \rangle\big] \\
&\leq \sqrt{\mathbb{E}_\pi\big[\big(\max_b |(\boldsymbol{\theta} - \boldsymbol{\theta}^*)^\top \nabla_{\boldsymbol{\theta}} f(\boldsymbol{\theta}_0; s', b)|\big)^2\big]}\sqrt{\mathbb{E}_\pi\big[\big(\nabla_{\boldsymbol{\theta}} f(\boldsymbol{\theta}_0; s, a)^\top (\boldsymbol{\theta} - \boldsymbol{\theta}^*)\big)^2\big]} \\
&= m\|\boldsymbol{\theta} - \boldsymbol{\theta}^*\|_{\boldsymbol{\Sigma}_\pi^*(\boldsymbol{\theta} - \boldsymbol{\theta}^*)}\|\boldsymbol{\theta} - \boldsymbol{\theta}^*\|_{\boldsymbol{\Sigma}_\pi},
\end{aligned}
$$

where we used the fact that $\boldsymbol{\Sigma}_\pi^*(\boldsymbol{\theta} - \boldsymbol{\theta}^*) = 1/m\mathbb{E}_\mu[\nabla_{\boldsymbol{\theta}} f(\boldsymbol{\theta}_0; s, \widetilde{b}_{\max})\nabla_{\boldsymbol{\theta}} f(\boldsymbol{\theta}_0; s, \widetilde{b}_{\max})^\top]$ and $\widetilde{b}_{\max} = \operatorname{argmax}_{b \in \mathcal{A}} |\langle \nabla_{\boldsymbol{\theta}} f(\boldsymbol{\theta}_0; s, b), \boldsymbol{\theta} - \boldsymbol{\theta}^* \rangle|$ according to (5.6). Substituting the above results into (B.6), we obtain

$$
\begin{aligned}
&\mathbb{E}_\pi\big[\big(\max_{b \in \mathcal{A}} \widehat{f}(\boldsymbol{\theta}; s', b) - \max_{b \in \mathcal{A}} \widehat{f}(\boldsymbol{\theta}^*; s', b)\big)\langle \nabla_{\boldsymbol{\theta}} f(\boldsymbol{\theta}_0; s, a), \boldsymbol{\theta} - \boldsymbol{\theta}^* \rangle\big] \\
&\leq m\|\boldsymbol{\theta} - \boldsymbol{\theta}^*\|_{\boldsymbol{\Sigma}_\pi^*(\boldsymbol{\theta} - \boldsymbol{\theta}^*)}\|\boldsymbol{\theta} - \boldsymbol{\theta}^*\|_{\boldsymbol{\Sigma}_\pi},
\end{aligned}
$$

which immediately implies

$$
\begin{aligned}
\langle \overline{\mathbf{m}}(\boldsymbol{\theta}) - \overline{\mathbf{m}}(\boldsymbol{\theta}^*), \boldsymbol{\theta} - \boldsymbol{\theta}^* \rangle &\geq m\|\boldsymbol{\theta} - \boldsymbol{\theta}^*\|_{\boldsymbol{\Sigma}_\pi} \cdot \big(\|\boldsymbol{\theta} - \boldsymbol{\theta}^*\|_{\boldsymbol{\Sigma}_\pi} - \|\boldsymbol{\theta} - \boldsymbol{\theta}^*\|_{\boldsymbol{\Sigma}_\pi^*(\boldsymbol{\theta} - \boldsymbol{\theta}^*)}\big) \\
&= m\|\boldsymbol{\theta} - \boldsymbol{\theta}^*\|_{\boldsymbol{\Sigma}_\pi} \cdot \frac{\|\boldsymbol{\theta} - \boldsymbol{\theta}^*\|_{\boldsymbol{\Sigma}_\pi}^2 - \gamma^2\|\boldsymbol{\theta} - \boldsymbol{\theta}^*\|_{\boldsymbol{\Sigma}_\pi^*(\boldsymbol{\theta} - \boldsymbol{\theta}^*)}^2}{\|\boldsymbol{\theta} - \boldsymbol{\theta}^*\|_{\boldsymbol{\Sigma}_\pi} + \gamma\|\boldsymbol{\theta} - \boldsymbol{\theta}^*\|_{\boldsymbol{\Sigma}_\pi^*(\boldsymbol{\theta} - \boldsymbol{\theta}^*)}} \\
&\geq m(1 - \alpha^{-1/2})\|\boldsymbol{\theta} - \boldsymbol{\theta}^*\|_{\boldsymbol{\Sigma}_\pi}^2 \\
&= (1 - \alpha^{-1/2})\mathbb{E}\big[\big(\widehat{f}(\boldsymbol{\theta}) - \widehat{f}(\boldsymbol{\theta}^*)\big)^2|\boldsymbol{\theta}_0\big],
\end{aligned}
$$

where the second inequality is due to Assumption 5.3 and the last equation is due to (B.5) and the definition of $\boldsymbol{\Sigma}_\pi$ in (5.5). $\qquad\square$

