# OpenReview forum: "A Finite-Time Analysis of  Q-Learning with Neural Network Function Approximation"
_ICLR.cc/2020/Conference — Reject_

### Official Review · AnonReviewer2 · 2019-10-22
**Official Blind Review #2**

**Rating:** 3

**Review:**

[Summary]
This paper studies the convergence of Q-Learning when a wide multi-layer network in the Neural Tangent Kernel (NTK) regime is used as the function approximator. Concretely, it shows that Q-learning converges with rate O(1/T) with data sampled from a single trajectory (non-i.i.d.) of an infinite-horizon MDP + a certain (stationary) exploration policy.

[Pros]
The results in this paper improve upon recent work on the same topic. It is able to handle multi-layer neural nets as opposed to two-layer, prove a faster rate O(1/T), and handle non-iid data (as opposed to iid data where (s,a,r,s’) are sampled freshly from the beginning at each step.)

The paper is generally well-written. The results and proof sketches are well presented and easy to follow. The proof seems correct to me from my check, including the indicator issue pointed out in the comments which I think can be easily fixed (by explicitly writing out the indicator and thus the Cauchy-Schwarz will still apply.)

[Cons]
The result in this paper seems more or less like a direct combination of existing techniques, and thus may be limited in bringing in new techniques / messages. Key technical bottlenecks that are assumed out in prior work are still assumed out in this paper with potentially different forms but essentially the same thing.

More concretely, the proof of the main theorem (Thm 5.4) seems to rely critically on Lemmas 6.2 and 6.3, both of which are rather straightforward adaptations of prior work:

Lemma 6.2 (concentration of stochastic gradients on linearized problem): Seems to me like almost the same as [Bhandari et al. 2018], expect that now the network is an affine function---rather than a linear function---of \theta, where the additional constant term f(\theta_0; s, a) depends on (s, a).

Lemma 6.3 (good landscape of linearized problem): Comparing with prior work (Theorem 6.3, Cai et al. 2019), this Lemma works by directly assuming out the property of the arg-max operator in Assumption 5.3, which has a slightly different form from, but is essentially the same thing as (Assumption 6.1, Cai et al. 2019).

To be fair, the paper has to deal with the linearization error of a multi-layer net, which is dealt with in Lemma 6.1 and should be valued. But still I tend to think the above adaptations are rather straightforward and technically not quite novel.

[Potential improvements]
I would like to hear more from the authors about the technical novelty in this paper, specifically how Lemma 6.1 - 6.3 compare with prior work. I would be willing to improve my evaluation if this can be addressed.


**Experience Assessment:**

I have read many papers in this area.

**Review Assessment: Checking Correctness Of Derivations And Theory:**

I carefully checked the derivations and theory.

**Review Assessment: Checking Correctness Of Experiments:**

N/A

**Review Assessment: Thoroughness In Paper Reading:**

I read the paper at least twice and used my best judgement in assessing the paper.

---

> ### Author Response · Authors · 2019-11-11
> **Response to official blind review #2**
>
> Thank you for your insightful comments. We would like to clarify that our paper is not a direct combination of existing results. We highlight that our main contributions are to provide (1) the first finite-time analysis of Q-learning with multi-layer neural network function approximation, and (2) the first finite-time analysis of neural Q-learning with non-i.i.d. data assumption. We agree with the reviewer's comment that our analysis is built on previous work of Bhandari et al. (2018) and Cai et al. (2019). However, the analysis for deep Q-learning with non-i.i.d. data is by no means trivial, as is shown in our following responses to your comments on each technical lemma.
>
> Overall, our proof of Theorem 5.4 was decomposed into three parts (the bounds of terms $I_1$, $I_2$ and $I_3$ in equation (6.5) of our paper), which are bounded using Lemmas 6.1, 6.2 and 6.3 respectively.
>
> In Lemma 6.1, we upper bound the difference between $\mathbf{g}_t$ and $\mathbf{m}_t$, where $\mathbf{g}_t$ is defined based on the Bellman residual error and the multi-layer neural network function, and $\mathbf{m}_t$ is defined on the same Bellman residual error but with a linearized function at the initial point $\theta_0$. This lemma requires a careful calculation of the bound on the temporal difference $\Delta_t(s_t,a_t,s_{t+1};\theta_t)$ and the linearization error, which is not presented in previous work.
>
> In Lemma 6.2, we characterize the bias of the stochastic gradient $\mathbf{m}_t(\cdot)$ and its idealized version $\overline{\mathbf{m}}(\cdot)$ whose definition does not depend on the Markov data trajectory. As we mentioned at the beginning of the proof of Lemma 6.2, our proof was indeed adapted from that in Bhandari et al. (2018). However, there are a few differences between their proof and ours. First, $\mathbf{m}_t(\cdot)$ and $\overline{\mathbf{m}}(\cdot)$ in our paper are defined based on a neural network function and its gradient, and thus the Lipschitz condition and the gradient norm bound are not trivial to derive. Second, the proof in Bhandari et al. (2018) is for TD learning, which does not directly apply to neural Q-learning.
>
> In Lemma 6.3, we used a slightly different assumption (Assumption 5.3 in the revision) from that of Cai et al. (2018). It is worth noting that our Assumption 5.3 follows the same idea of Melo et al. (2008), Zou et al. (2019) and Chen et al. (2019), which can be interpreted as the advantage of the greedy policy over the learning policy. In contrast, Assumption 6.1 in Cai et al. (2019) directly imposes the condition on the difference between action value functions at two different policies. Therefore, our proof is based on bounding the eigenvalue of the difference between two covariance matrices (i.e., $\hat\Sigma_{\pi}$ and $\hat\Sigma_{\pi}^*(\theta)$), which is different from that of Cai et al. (2019).

---

### Official Review · AnonReviewer1 · 2019-10-23
**Official Blind Review #1**

**Rating:** 6

**Review:**

This paper provides a finite-time analysis of a neural Q-learning algorithm, where the data are generated from a Markov decision process and the action-value function is approximated by a deep ReLU neural network. When the neural function is sufficiently over-parameterized, the O(1/T) convergence rate is attained.

Pros: This paper makes theoretical contribution to the understanding of neural Q-learning. This is an important but difficult task. The recent finite-time analysis on Q-learning either assumes a linear function approximation or an i.i.d. setting in the neural Q-learning. This paper makes a first attempt to study the neural Q-learning with Markovian noise. Overall, this paper is very easy to follow.

Cons: In spite of its theoretical contributions, this paper has a few major issues.

1. The projection step relies on a parameter $\omega$ which is unknown in practice. In theorem, $\omega = C m^{-1/2}$ for some unknown constant $C$. It would be of practical interests to seek other proof techniques to avoid such projection step. For instance, Srikant and Yang (2019) and Chen et al. (2019) removed this projection step in the finite-time analysis of Q-learning with a linear function approximation.

2. Assumption 5.3 is problematic for the considered neural Q-learning setting. The matrix $\hat{\Sigma}_{\pi}$ is of a very large dimension in the order of $O(m^2) * O(m^2)$ where the width of the neural network $m$ is assumed to diverge in the Theorem 5.4 for the over-parameterization purpose. Given the diverging dimension scenario, it is problematic to ensure Assumption 5.3. Moreover, it is unclear how to verify this condition in practice. In the literature, Melo et al. (2008) and Zou et al. (2019b) assumed a similar condition, which is OK because in the Q-learning with linear function approximation, this matrix reduces to the covariance matrix of the feature vector.

3. The error rate in Theorem 5.4 is an increasing function of the layer $L$ in DNN, which is counterintuitive. A typically practical observation is that a larger $L$ is better.




**Experience Assessment:**

I have read many papers in this area.

**Review Assessment: Checking Correctness Of Derivations And Theory:**

I assessed the sensibility of the derivations and theory.

**Review Assessment: Checking Correctness Of Experiments:**

N/A

**Review Assessment: Thoroughness In Paper Reading:**

I read the paper at least twice and used my best judgement in assessing the paper.

---

> ### Author Response · Authors · 2019-11-11
> **Response to official blind review #1**
>
> Thank you for your helpful comments. We address them point by point as follows.
>
> Q1: "The projection step relies on a parameter $\omega$ which is unknown in practice. It would be of practical interests to seek other proof techniques to avoid such projection step. ..."
>
> A1: The unknown constant $C$ is often treated as a hyperparameter and can be tuned using grid search in practice. We agree that it would be interesting to explore the possibility of removing the projection step as what Srikant and Ying (2019) and Chen et al. (2019) did in the linear approximation setting. We have discussed their methods in the related work section. However, adapting their proof techniques would completely change our algorithm and our current analysis framework. So we will investigate the projection-free version of our algorithm in the future work.
>
> Q2: "Assumption 5.3 is problematic for the considered neural Q-learning setting. The matrix $\hat{\Sigma}_{\pi}$ is of a very large dimension in the order of $O(m^2) * O(m^2)$ where the width of the neural network $m$ is assumed to diverge in Theorem 5.4 for the over-parameterization purpose. ..."
>
> A2: In our previous submission, we did not require $m$ to go to infinity and thus the matrix $\hat\Sigma_{\pi}$ is well defined and positive definite.  However, the minimum eigenvalue of $\hat\Sigma_{\pi}$ could be very small and hence $\alpha$ (the minimum eigenvalue of $\hat\Sigma_{\pi}-\gamma^2\hat\Sigma_{\pi}^*(\theta)$) would be a very small quantity which can slow down the convergence rate. Based on this observation, we agree that the previous assumption is too restrictive and we have removed the assumption on the minimum eigenvalue in the revision. In particular, we relax the previous assumption to a much milder one where we only require the difference between the two matrices ($\hat\Sigma_{\pi}$ and $\hat\Sigma_{\pi}^*(\theta)$) to be positive definite (see Assumption 5.3 in the revision). Under this milder assumption, we proved that neural Q-learning converges with an $O(1/\sqrt{T})$ rate. This result matches the convergence rate of neural Q-learning in Cai et al. (2019) where only a two-layer neural network approximator is used and the data are assumed to be i.i.d. generated.
>
> Q3: "The error rate in Theorem 5.4 is an increasing function of the layer $L$ in DNN, which is counterintuitive. A typically practical observation is that a larger $L$ is better."
>
> A3: The dependence on $L$ in the error rate in Theorem 5.4 comes from Lemma 6.1 which characterizes the approximation error between the linearized gradient $\mathbf{m}_t$ and the gradient term $\mathbf{g}_t$. The dependency of $L$ can be removed by choosing a smaller $\omega=C_0m^{-1/2}L^{-9/4}$. Please see the updated Theorem 5.4 in the revision.

---

> > ### Comment · AnonReviewer1 · 2019-11-13
> > **Thanks for the revision**
> >
> > Thank the authors for addressing all my comments. I feel satisfied with all the revisions. Although the rate has been changed to $O(1/\sqrt{T})$, I still feel this paper makes a good theoretical contribution to neural Q-learning.

---

### Official Review · AnonReviewer3 · 2019-10-23
**Official Blind Review #3**

**Rating:** 6

**Review:**

This paper introduces a finite time analysis of Q-learning with neural network function approximators across multiple layers and no iid assumption.

[Pros]
+ Provides a novel way to analyze Q learning with nn function approximators that can be applied to other algorithms (notably in my mind, TD in actor critic where iid assumptions are often violated).

[Cons]
+ The novelty is a bit unclear other than the non-iid assumption. We note that modern Q-learning tends to use batching so doesn't require much of an iid assumption anyways, but this allows for more robust proofs in TD settings with non-iid training.
+ The paper was a bit dense and hard to follow, we suggest reducing p.8 to have more discussion with references to proofs in the Appendix as in Chen2019.
+ As the authors admit in open commentary, there is a mistake to be fixed which needs to be reviewed before acceptance. I think there is value to this work, however, would require seeing the change to assess a revision.


**Experience Assessment:**

I have read many papers in this area.

**Review Assessment: Checking Correctness Of Derivations And Theory:**

I assessed the sensibility of the derivations and theory.

**Review Assessment: Checking Correctness Of Experiments:**

N/A

**Review Assessment: Thoroughness In Paper Reading:**

I made a quick assessment of this paper.

---

> ### Author Response · Authors · 2019-11-11
> **Response to official blind review #3**
>
> Thank you for your constructive comments. We address your questions as follows.
>
> Q1: "The novelty is a bit unclear other than the non-iid assumption. We note that modern Q-learning tends to use batching so doesn't require much of an iid assumption anyways, but this allows for more robust proofs in TD settings with non-iid training."
>
> A1: Existing work on neural Q-learning (Cai et al, 2019) requires to resample a new pair of data $(s, a, s')$ at every iteration from the initial data distribution. This is not efficient in practice since one step along the trajectory may not give a good prediction of the policy. In contrast, our paper study the case where data $(s_t,a_t,s_{t+1})$ is drawn from a consecutive trajectory generated by the learning policy. Apart from the non-i.i.d. data generation, another contribution of our paper is to study the convergence of Q-learning with multi-layer neural network approximation. The extension from the two layer case in Cai et al. (2019) to our multi-layer case is not easy since the linearization error can not be calculated directly.
>
> Q2: "The paper was a bit dense and hard to follow, we suggest reducing p.8 to have more discussion with references to proofs in the Appendix as in Chen2019."
>
> A2: Thank you for the suggestion. We have added additional discussions and more details of the proof in Section 6 to make the proof easier to follow. In order to make the proof coherent, we did not divide the proof into several parts and move some parts of the proof to the appendix. Please let us know if you have any further suggestion.
>
> Q3: "As the authors admit in open commentary, there is a mistake to be fixed which needs to be reviewed before acceptance. I think there is value to this work, however, would require seeing the change to assess a revision."
>
> A3: We have fixed the problem of the indicator function used in the proof of Lemma 6.3. In particular, we chose to modify the definition of $b_{\max}$ in (5.6) to be $b_{\max}(\theta)=\arg\max_{b\in\mathcal{A}}|\langle\nabla_{\theta} f(\theta;s,b),\theta\rangle|$ which is similar to the definition used in Chen et al. (2019) (Note that their paper is for linear function approximation and thus $b_{\max}(\theta)=\arg\max_{b\in\mathcal{A}}|\phi(s,b)^{\top}\theta|$). This does not change the result of Lemma 6.3. See page 5 and pages 17-18 of the revision for the details.

---

### Public Comment · ~Sussard_Julard1 · 2019-10-08
**Missing related work**

This paper uses linear approximation of neural networks, which borrows the analysis using Neural tangent kernels (NTK). There are a lot of papers on NTK recently and are neglected by the author, including the paper introduced NTK. Please cite at least the following papers:

1. Jacot et al. Neural Tangent Kernel: Convergence and Generalization in Neural Networks
2. Allen-Zhu et. al. Learning and generalization in overparameterized neural networks, going beyond two layers
3. Arora et. al. Fine-grained analysis of optimization and generalization for overparameterized two-layer neural networks
4. Cai et. al. A gram-gauss-newton method learning overparameterized deep neural networks for regression problems
5. Chizat et. al. On the global convergence of gradient descent for over-parameterized models using optimal transport

---

> ### Author Response · Authors · 2019-10-11
> **Re: Missing related work.**
>
> Thank you for pointing out the relevant papers on neural tangent kernels. We will cite them in the revision (during the author response phase when we can update the submission file).

---

### Public Comment · ~Sussard_Julard1 · 2019-10-08
**Potentially contains error in the anlysis. Remarks on the contribution**

1. This work combines the ideas of the following four papers:  (1): [Cao2019] for handing deep overparameterized NN using neural tangent kernel, (2) [Bhandari2018] for the finite-time analysis of temporal difference learning with non-iid data; (3) [Cai2019] for temporal-difference learning with 2-layer overparametrized NN, and (4) [Melo2007] for linear Q-learning.

The general analysis framework follows from [Bhandari2018], while the technical assumption for Q-learning (Assumption 5.3) is borrowed from [Melo2007], which is also used in [Zou2019]. Specifically, the main theorems, Theorems 5.4 and 5.6, have similar counterparts in [Cai2019], albeit for 2-layer NN. Compared with that work, this work has 3 main differences/improvements: (1) handling deep nets and (2) non i.i.d. data and (3) use a different assumption for Q learning, namely Assumption 5.3.  Theorem 5.4 is depends on Lemmas 6.1, 6.2, 6.3, and the proof strategy for this theorem is similar to [Bhandari2018] and [Cai2019]. Lemma 6.1 is adapted from [Cao2019], Lemma 6.2 is borrowed from [Bhandari2018] to handle non-iid data, and Lemma 6.3 is adapted from [Zou2019]. Thus, it seems that this work is a combination of existing results.


More importantly, the proof of Lemma 6.3, which is based on Assumption 5.3, might not be correct. Specifically, on page 16, after getting (B.5), the authors separate two cases depending on whether $< \nabla _{\theta} f(\theta_0, s, a) , \theta - \theta ^*> $ is positive or not. In each case, the authors establish an upper bound using Cauchy-Schwarz. This is exactly the same as the proof of Theorem 1 in [Melo2007] (http://icml2008.cs.helsinki.fi/papers/652.pdf). However, due to the indicator functions, you cannot directly combine these two cases. The indicator functions should be taken into consideration. Moreover, in recent work on linear Q-learning, [Chen 2019] (which is not cited), the authors state that the proof in [Melo2007] is NOT CORRECT! Thus, by directly following [Melo2007]'s proof, the proof of Lemma 6.4 is not correct.

[Chen2019] gives the following remarks (https://arxiv.org/pdf/1905.11425v1.pdf):
``"One such condition was proposed in [Melo2007] to restrict the sampling policy to be close enough to the optimal policy. However, we could not verify the correctness of the proof of the main theorem (Theorem 1) in [Melo2007] even after personal communication with the corresponding author"

Thus, it would be great if the authors could check the proof and resolve the technical issue inherit from [Melo2007].


Cao2019: Generalization Bounds of Stochastic Gradient Descent for Wide and Deep Neural Networks
Bhandari2018: A Finite-Time Analysis of Temporal Difference Learning With Linear Function Approximation
Cai2019; Neural Temporal-Difference Learning Converges to Global Optima
Melo2007: An Analysis of Reinforcement Learning with Function Approximation
Zou2019: Finite-sample analysis for sarsa with linear function approximation
Chen2019: Performance of Q-learning with Linear Function Approximation: Stability and Finite-Time Analysis

---

> ### Author Response · Authors · 2019-10-11
> **Re: Potentially contains error in the analysis. Remarks on the contribution.**
>
> 1. The additional reference: We are happy to cite and comment the recent arXiv paper you mentioned.
>
> 2. Contribution: we have emphasized our contributions in the abstract and introduction of our paper. As we displayed in Table 1 in our paper, our work is different from and superior to existing papers in multiple ways. To summarize, this is the first work that studies the convergence of Q-learning with deep neural network function approximation. Compared with existing work [Cai2019] which studies Q-learning with two-layer neural network function approximation with i.i.d. noise assumptions, we study the Markovian noise of deep Q-learning. Our convergence rate is also sharper than that of existing work for two-layer neural networks even though our setting is much more challenging.
>
> 3. Correctness: we clarify the “correctness” you suspected as follows:
>
> (1) The indicator function: while our proof technique is similar to that of [Melo2007], we did not explicitly use the indicator function in our proof as [Melo2007]. Similar results can also be found in Theorem 5.1 and Lemma 5.3 in the reference [Chen2019] (https://arxiv.org/pdf/1905.11425.pdf) pointed out by you, under a slightly different assumption. We will discuss this in our revision during the author response phase.
>
> (2) The “error” in [Melo2007]: the arXiv paper [Chen2019] (https://arxiv.org/pdf/1905.11425.pdf) pointed out by you has removed the comment “However, we could not verify the correctness of the proof of the main theorem (Theorem 1) in [Melo2007] even after personal communication with the corresponding author” in their second version. We don’t find any unfixable error in [Melo2007].

---

> ### Public Comment · ~Sussard_Julard1 · 2019-10-15
> **Re: Re: Potentially contains error in the analysis. Remarks on the contribution.**
>
> Thanks for your reply. In terms of the correctness of the proof, I am still unconvinced. The reason is that currently your proof separates two cases depending on $\langle \nabla_{\theta} f_{\theta} (\theta_0; s,a), \theta - \theta_0 \rangle $ is positive or negative. Note that this is a random variable.  Thus, your proof seems essentially the same as having an indicator. More specifically, if this random variable is nonnegative, we write $\mathrm{1}_{E} = \{\langle \nabla_{\theta} f_{\theta} (\theta_0; s,a), \theta - \theta_0 \rangle \geq 0 \}$, then the equation below (B.5) is the same as saying
> 	$$
> 	\mathbb{E}_{\pi}\left[\left(\widehat{f}\left({\theta} ; s^{\prime}, b_{\max }\right)-\widehat{f}\left({\theta}^{*} ; s^{\prime}, b_{\max }^{*}\right)\right)\left\langle\nabla_{{\theta}} f\left({\theta}_{0} ; s, a\right), {\theta}-{\theta}^{*}\right\rangle \cdot \mathrm{1}_{E} \right] \leq \mathbb{E}_{\pi}\left[\left({\theta}-{\theta}^{*}\right)^{\top} \nabla_{{\theta}} f\left({\theta}_{0} ; s^{\prime}, b_{\max }\right) \nabla_{{\theta}} f\left({\theta}_{0} ; s, a\right)^{\top}\left({\theta}-{\theta}^{*}\right) \cdot \mathrm{1}_{E}\right].
> $$
> Then, in the Cauchy-Schwarz inequlity afterwards, you need to handle this indicator function directly because you take expectations when using Cauchy-Schwarz.
>
> In a word, you cannot separate the random variable into different cases and compute expectations of its functions on each difference cases separately!

---

> > ### Author Response · Authors · 2019-10-18
> > **As stated in our last response, this minor issue is easily fixable.**
> >
> > As we stated in the last response to you, the minor issue you mentioned is easy to fix, and we will update the revision when the author response phase starts. To give you a general idea, there are at least two ways to fix it. One way is to follow our current proof, which is similar to the proof of Theorem 1 in [Melo2007], and change the $\max$ operator in the penultimate equation of our proof for Lemma 6.3 to a summation operator, which leads to a $\sqrt{2}$ factor on the right hand side of this inequality. This will introduce a factor of 2 in front of $\gamma^2$ in Assumption 5.3. Another way of fixing it is to impose a slightly different assumption on the learning policy like in [Chen2019]. In particular, if we change the definition of $b_{\max}(\theta)$ used in equation (5.6) to be $b_{\max}(\theta)=\arg\max_{b\in\mathcal{A}}|\langle\nabla_{\theta} \hat f(\theta;s,b),\theta\rangle|$, then the same result holds under our current Assumption 5.3. Neither of these fixes will affect the conclusion of our paper. We will elaborate this in detail in our revision.

---

> > > ### Public Comment · ~Sussard_Julard1 · 2019-10-18
> > > **Re: As stated in our last response, this minor issue is easily fixable.**
> > >
> > > Thanks for your reply. But I don't think the issue can be easily fixed by changing max into a sum. Otherwise, [Chen2019] would have done so instead of constructing a new assumption. I would admit that the issue can be successfully solved following the proof in [Chen2019] though.

---

### Public Comment · ~Matt_Theodore1 · 2019-10-18
**Assumption 5.3 seems unattainable. The hessian matrix in a overparametrized NN cannot have lower bounded smallest eigenvalue.**

Assumption 5.3 assumes that $\hat \Sigma_{\pi}$ defined in equation 5.5 has eigenvalues lower bounded by $\alpha$, which seems a critical condition which leads to the $O(1/T)$ convergence rate.

However, this assumption cannot be true.  As shown in (5.5) and assumption 5.3, we have
$$
\hat{\mathbf{\Sigma}}_{\pi}=1 / m \mathbb{E}_{\pi}\left[\nabla_{{\theta}} \widehat{f}({\theta} ; s, a) \nabla_{{\theta}} \widehat{f}({\theta} ; s, a)^{\top}\right] \geq \alpha \cdot I_{m},
$$
where $I_m$ is the identity matrix in $R^m$, $\alpha$ is a fixed constant, and $m$ is the total number of parameters. Thus, the trace of $\hat{\mathbf{\Sigma}}_{\pi}$ is $\Omega(m)$.

However, as shown in [Jacot et al] or [Du et al], the Gram matrix defined as
$$
\mathbf{G} =  1 / m \mathbb{E}_{\pi}\left[\nabla_{{\theta}}^{\top} \widehat{f}({\theta} ; s, a) \nabla_{{\theta}} \widehat{f}({\theta} ; s, a)\right]  ,
$$
has trace bounded by $O(n)$.

With some simple linear algebra we have
$$
\Omega(m) = \textrm{trace} (\hat{\mathbf{\Sigma}}_{\pi} ) = \textrm{trace}(\mathbf{G} ) = O(n),
$$
which contradicts with the fact that $m $ is much larger than $n$, as assumed in the overparametrization setting.

---

> ### Author Response · Authors · 2019-10-20
> **Re: Assumption 5.3 seems unattainable. The hessian matrix in a overparametrized NN cannot have lower bounded smallest eigenvalue.**
>
> We would like to clarify that you had misunderstood the definitions of $\mathbf{G}$ in [Jacot et al, Du et al] and $\widehat\Sigma_{\pi}$ in our paper, which are totally different matrices. It should be noted that $\widehat\Sigma_{\pi}$ is NOT the Gram matrix defined in [Jacot et al., 2018] or [Du et al., 2019]. We explain the definition of $\widehat\Sigma_{\pi}$ as follows.
>
> According to (4.2) in our paper, we have that $\mathbf{\theta}\in\mathbb{R}^{m^2(L-1)+m(d+1)}$ is the concatenation of vectorized weight matrices, which means the gradient $\nabla_{\mathbf{\theta}}\widehat f(\mathbf{\theta};s,a)$ is also a $p=m^2(L-1)+m(d+1)$ dimensional vector. The matrix $\widehat\Sigma_{\pi}$ used in Assumption 5.3 is defined as
> $$
> \widehat\Sigma_{\pi}=1/m\mathbb{E}_{\pi}\big[\nabla_{\mathbf{\theta}}\widehat f(\mathbf{\theta};s,a)\nabla_{\mathbf{\theta}}\widehat f(\mathbf{\theta};s,a)^{\top}\big],
> $$
> which is a $p\times p$ matrix, and the expectation $\mathbb{E}[\cdot]$ is taken over the data distribution in the feature space of the state-action pair $(s,a)$. The expectation $\mathbb{E}[\cdot]$ over data distribution means $\widehat\Sigma_{\pi}$ is defined based on an infinite number of data points. $\widehat\Sigma_{\pi}$ has nothing to do with the overparameterization.  It can be positive definite under certain conditions on the learning policy and the transition probability kernel, which is a standard assumption in the literature [Melo et al., 2008;Bhandari et al., 2018; Zou et al., 2019].
>
> For the Gram matrix $\mathbf{G}$, we first point out that your definition is incorrect. According to [Jacot et al., 2018] and [Du et al., 2019], for an $L$-layer deep neural network, the Gram matrix (at the initial point $\mathbf{\theta}_0$) is denoted as $\mathbf{G}^{(L)}(0)\in\mathbb{R}^{n\times n}$, which is defined based on $n$ data points $\{(s_i,a_i)\}_{i=1}^{n}$. Specifically, for any $j,k=1,\ldots,n$, the $(j,k)$-th entry of the Gram matrix in their paper is defined as
> $$
> G_{jk}^{(L)}(0)=\nabla_{\mathbf{\theta}} \widehat f(\mathbf{\theta}_0;s_{j},a_{j})^{\top}\nabla_{\mathbf{\theta}} \widehat f(\mathbf{\theta}_0;s_{k},a_{k}).
> $$
> Therefore, the definition of the Gram matrix depends on a fixed number of data points with size $n$.
>
> To summarize, the Gram matrix is a $n\times n$ matrix defined based on $n$ data points.
> It is clear that the matrix $\widehat\Sigma_{\pi}\in\mathbb{R}^{p\times p}$ in our paper is not the Gram matrix and is independent of $n$.

---

> > ### Public Comment · ~Matt_Theodore1 · 2019-10-22
> > **The matrix $\hat \Sigma_{\pi}$ has diverging trace**
> >
> > In your work, as $\hat \Sigma_{\pi}$ is a $p\times p$ matrix and its eigenvalues are lower bounded, its trace will be $\Omega(p) = \Omega(m^2) $.  That means, as $m$ goes to infinity, the trace of this matrix will blow up. This seems very strange as estimating this matrix would be impossible using $n$ data points.
> > How does your analysis cope with the fact that $\hat \Sigma_{\pi}$ is divergent as $m$ goes to infinity?

---

> > > ### Author Response · Authors · 2019-10-24
> > > **Re: The matrix $\hat \Sigma_{\pi}$ has diverging trace**
> > >
> > > You still have a misunderstanding of our assumption and analysis. In our paper, we do not need to estimate the population matrix $\hat\Sigma_{\pi}$ based on $n$ data points. Assumption 5.3 is imposed on $\hat\Sigma_{\pi}$ which by definition already takes the expectation over the data distribution (according to our previous response, this means infinite data points). This assumption is used in the proof of Lemma 6.3. According to our Theorems 5.4 and 5.6, we do not require $m$ to go to infinity, which is the same as some recent papers on NTK such as [Allen-Zhu et al., 2019, Du et al., 2019, Zou et al., 2019].

---

> > > > ### Public Comment · ~Matt_Theodore1 · 2019-10-26
> > > > **Re: Re: The matrix $\hat \Sigma_\pi$  has diverging trace**
> > > >
> > > > Thanks for your reply. The reason I say $m$ is allowed to go to infinity is to say that $m$ is much larger than $n$. For example, in the NTK paper, they typically assume that $m  = \Omega(n^8)$.
> > > >
> > > > However, the contradiction in your paper is just there. I agree with you that $\Sigma_{\pi}$ is a population matrix. Do you agree that its trace is $\Omega(p) = \Omega(m^2)$? That is  $\| \hat\Sigma_{\pi} \|_{\textrm{fro}} = \Omega(m^2)$. However, as shown in your Lemma B.1, for all $x$,  you have shown $\| \nabla_{\theta} f (\theta; x) \|_2 \leq C\cdot \sqrt{m}$ for some constant $C$. This means that $$\hat \Sigma_{\pi} = 1/ m \cdot \mathbb{E}_{x \sim \pi} [  \nabla_{\theta} f (\theta; x)\nabla_{\theta} f (\theta; x)^\top ] $$
> > > >  has Frobenious norm bounded by $C^2$, as $$\| \nabla_{\theta} f (\theta; x)\nabla_{\theta} f (\theta; x)^\top \|_{\textrm{fro} } = \| \nabla_{\theta} f (\theta; x) \|_2 ^2 \leq C^2 \cdot m.$$
> > > >  By your Assumption 5.3 and Lemma B.1,
> > > > $$
> > > > \Omega(m^2)  = \textrm{trace} (\hat \Sigma_{\pi})  \leq C^2.
> > > > $$
> > > > This must be wrong!

---

### Author Response · Authors · 2019-11-11
**Response to all reviewers: major changes in the revision**

Thank you for reviewing our submission. We have addressed all your questions in the individual responses to each reviewer. Here we summarize the main changes made in our revision for you to have a quick overview of it.

1. The first major revision is in Assumption 5.3. Previously, we require $\Sigma_{\pi}-\gamma^2\Sigma_{\pi}^*(\theta)\succ \alpha$ for a positive constant $\alpha$. However, during the discussion in one of the open commentary, we found that this assumption is restrictive and hard to be verified in practice. We thus relaxed this condition to only assume $\Sigma_{\pi}-\alpha\gamma^2\Sigma_{\pi}^*(\theta)\succ 0$ for some positive constant $\alpha$. Note that this assumption is much milder and easy to be attained since both $\Sigma_{\pi}$ and $\Sigma_{\pi}^*(\theta)$ are positive definite by definition. The high level idea of the assumption remains the same, which ensures that the learning policy is not too much worse than the greedy policy. Following this milder assumption, we proved an $O(1/\sqrt{T})$ convergence rate of Q-learning  with multi-layer neural network function approximation under non-i.i.d. data. Consequently, we have modified the corresponding statement in the introduction, the rate in Table 1, and the result in Theorems 4.5. and 4.6.

2. We changed the definition of $b_{\max}$ in (5.6) to be $b_{\max}$ in (5.6) to be $b_{\max}(\theta)=\arg\max_{b\in\mathcal{A}}|\langle\nabla_{\theta} f(\theta;s,b),\theta\rangle|$. This is consistent with Chen et al. (2019). This does not affect the result of Lemma 6.3.

---

### Decision · Program_Chairs · 2019-12-19

**Decision:**

Reject

**Comment:**

This was an extremely difficult paper to decide, as it attracted significant commentary (and controversy) that led to non-trivial corrections in the results.  One of the main criticisms is that the work is an incremental combination of existing results.  A potentially bigger concern is that of correctness: the main convergence rate was changed from 1/T to 1/sqrt{T} during the rebuttal and revision process.  Such a change is not trivial and essentially proves the initial submission was incorrect.  In general, it is not prudent to accept a hastily revised theory paper without a proper assessment of correctness in its modified form.  Therefore, I think it would be premature to accept this paper without a full review cycle that assessed the revised form.  There also appear to be technical challenges from the discussion that remain unaddressed.  Any resubmission will also have to highlight significance and make a stronger case for the novelty of the results.